# Vegetation Greening Weakened the Capacity of Water Supply to China's South-North Water Diversion Project

Jiehao Zhang[1,2], Yulong Zhang[3], Ge Sun[4], Conghe Song[2], Matthew P. Dannenberg[5], Jiangfeng Li[1], Ning Liu[4], Kerong Zhang[6], Quanfa Zhang[6], Lu Hao[7]

[1]Department of Land Management, China University of Geosciences, Wuhan, 430074, China.
[2]Department of Geography, University of North Carolina at Chapel Hill, Chapel Hill, NC 27599, USA.
[3]Institute for a Secure and Sustainable Environment, University of Tennessee, Knoxville, TN 37902, USA.
[4]Eastern Forest Environmental Threat Assessment Center, Southern Research Station, USDA Forest Service, Research Triangle Park, NC 27709, USA.
[5]Department of Geographical and Sustainability Sciences, University of Iowa, Iowa City, IA 52242, USA.
[6]Wuhan Botanical Garden, Chinese Academy of Sciences, Wuhan, 430074, China.
[7]Key laboratory of Meteorological Disaster, Ministry of Education (KLME)/Jiangsu Key Laboratory of Agricultural Meteorology, Nanjing University of Information Science and Technology, Nanjing, 210044, China.

Corresponding to: Conghe Song (csong@email.unc.edu); Jiangfeng Li (jfli0524@163.com)

**Abstract.** Recent climate change and vegetation greening have important implications for global terrestrial hydrological cycles and other ecosystem functions, raising concerns about the watershed water supply capacity for large water diversion projects. To address this emerging concern, we built a hybrid model based on the Coupled Carbon and Water (CCW) and Water Supply Stress Index (WaSSI) models and conducted a case study on the Upper Han River Basin (UHRB) in central China that serves as the water source area to the middle route of the South-North Water Diversion Project (SNWDP). Significant vegetation greening occurred in the UHRB during 2001-2018, largely driven by the widespread afforestation in the region, with the normalized difference vegetation index increasing at a rate of 0.5±0.1 % yr$^{-1}$ ($p < 0.05$) but no significant trends in climate during the same period (albeit with large interannual variability). Annual water yield greatly decreased, and vegetation greening alone induced a significant decrease in water yield of 3.2±1.0 mm yr$^{-1}$ ($p < 0.05$). Vegetation greening could potentially reduce the annual water supply by 7.3 km$^3$ on average, accounting for 77% of the intended annual water diversion volume of SNWDP. Although vegetation greening can bring enormous ecosystem goods and services (e.g., carbon sequestration & water quality improvement), it could aggravate the severity of hydrological drought. Our analysis indicated that vegetation greening in UHRB reduced about a quarter of water yield on average during drought periods. Given the future warming and drying climate is likely to continue to raise evaporative demand and exert stress on water availability, the potential water yield decline induced by vegetation greening revealed by our study need to be taken into account in the water resources management over the UHRB while reaping other benefits of forest protection and ecological restoration.

# 1 Introduction

As the world's population and economy expand under a changing climate, human demand for freshwater increases and water shortage has become a concern globally (Jackson et al., 2001). Water diversion with long-range transport has become an alternative measure to alleviate regional water shortage (Emanuel et al., 2015), but the sustainability of such projects depends on water supply from the donor watersheds. Watershed water supply is becoming increasingly uncertain due to rapid land use and land cover change and global climate change. In addition, vegetation greening has been observed globally in many regions as a combined result of climate change, land cover change (e.g., afforestation), and $CO_2$ fertilization (Chen et al., 2019; Guay et al., 2014; Zhang et al., 2017; Zhu et al., 2016). How this 'greening up' affects water yield in source regions of water diversion projects is unclear. A better understanding of the hydrological effects of vegetation greening on water supply of water diversion projects is critical for designing watershed management strategies to meet future water demand under a changing climate (van Loon et al., 2016).

The South to North Water Diversion Project (SNWDP) is the largest hydrological engineering project (in terms of investment) in the world to mitigate the water shortage in North China (Zhang, 2009). The Upper Han River Basin (UHRB), a subtropical basin in central China, is the water source area for the middle route of the SNWDP. Considering the importance of the UHRB for the SNWDP, China implemented large-scale afforestation and ecological restoration projects to safeguard water quality and increase soil water storage from the UHRB. The afforestation-driven greening of the UHRB has been larger and more significant than in most other parts of the world (Chen et al., 2019). The greening of the UHRB could create a trade-off between ecological restoration and water availability (Jackson et al., 2005). The increased forest cover in the UHRB reduced sediment in the streamflow and improve water quality (Li et al., 2008; Qi et al., 2019), and significantly increased carbon storage (Zhang et al. 2014). However, afforestation and vegetation greening could also exert considerable influences on the water cycle by increasing vegetation water use (Bai et al., 2019; Li et al., 2018). Specifically, enhanced vegetation activity from greening potentially consumes more water through transpiration, which could lead to a reduction in water yield (Bai et al., 2020; Cao et al., 2016; Li et al., 2018), especially during drought periods (Teuling et al., 2013; Tian et al., 2018).

Since the UHRB was chosen as the source water areas for the middle route of the SNWDP, whether the basin has the capacity to supply enough water to the project is one of the most debated issues about the project (Barnett et al., 2015; Stone, 2006; Zhang et al., 2020). The planned total water diversion each year during Phase I of the middle route project is 9.5 km$^3$, accounting for nearly one-third of the mean annual runoff of the UHRB. However, water yield in the UHRB has sharply declined since the early 1990s (Chen et al., 2007; Liu et al., 2012; She et al., 2017). Moreover, the UHRB is quite vulnerable to hydrological drought events (Xu et al., 2011; Zhang et al., 2018) because about 70% of its precipitation is recycled back to the atmosphere via evapotranspiration (ET) (China Meteorological Administration, 2019). The capacity of water supply to the SNWDP in drought years is only half of that of a normal year (Wang and Yang, 2005), exerting a large influence on the water supply capacity of the water diversion project, especially at the seasonal scale. The drought risks are likely compounded by warming-induced increases in evaporative demand due to increasing vapor pressure deficit (Cook et al., 2014, 2020; Lesk et

al., 2016; Williams et al., 2020). Combined with the effects of afforestation and vegetation greening, the uncertainties in water supply capacity to SNWDP are amplified. How the rapid and widespread greening has affected water yield in the UHRB, and thus the water supply to the middle route SNWDP, remains largely unknown.

The small paired-watershed experiment approach is the 'gold standard' for investigating the mechanisms of changes in water balance in response to vegetation (Bosch and Hewlett, 1982), but this method is not feasible for large basins (Wei et al., 2008). Eco-hydrological models based on remote sensing inputs provide an efficient way to understand hydrological processes at a high spatial resolution over large areas and long time periods (Wang and Dickinson, 2012). To investigate interactions among vegetation, climate, and the water cycle in the UHRB, a hydrological model should consider the effects of both vegetation and climate change and mechanistic understanding of hydrology. While many studies have investigated hydrological processes using mechanistic land surface models (Baker and Miller, 2013; Shin et al., 2019; Xi et al., 2018), their complex model structures and large number of parameters limit their applicability at a large spatial scale. Likewise, carbon and water fluxes of vegetation are tightly connected, but many hydrological models lack biological constraints of photosynthetic carbon uptake in quantifying hydrological variables.

To navigate this trade-off between mechanistic carbon-water linkage and computational efficiency, we developed and applied a new hybrid model that integrates two existing models: the 'water centric' Water Supply Stress Index (WaSSI) model (Sun et al., 2011; Caldwell et al., 2012; Liu et al., 2020) and the 'carbon centric' Coupled Carbon and Water (CCW) model (Zhang et al., 2016, 2019b). Here, we use this coupled model to address the following questions: 1) What were the spatial and temporal patterns of annual and monthly water yield (WY) in the UHRB from 2001 to 2018? 2) To what extent did the rapid local vegetation greening affect WY in the UHRB and thus water supply for the SNWDP? 3) How did the local vegetation greening change hydrological drought risks in the UHRB? Overall, our goal is to improve understanding of the effects of local vegetation greening on the water balance and hydrological drought and to provide a scientific basis for managing watersheds that serve as critical water supply in inter-basin water diversion projects.

## 2 Methods and data

### 2.1 Study area

The Han River in central China covers approximately $1.59 \times 10^5$ km$^2$ with a total length of 1,577 km (Jin and Guo, 1993; Yang et al., 1997), making it the longest tributary of the Yangtze River. Its mountainous upper reaches (31°20'–34°10' N, 106°–112° E; 210 – 3,500 m a.s.l) are 925 km long and drain an area of approximately $9.5 \times 10^4$ km$^2$ (Yang et al., 1997). The historical mean annual runoff of the UHRB is 41.1 km$^3$ (though with high interannual variability) (Yang et al., 1997). The Danjiangkou Reservoir, located in the easternmost tip of the UHRB, stores runoff from the UHRB and serves as the water source for the middle route of the SNWDP (Figure 1).

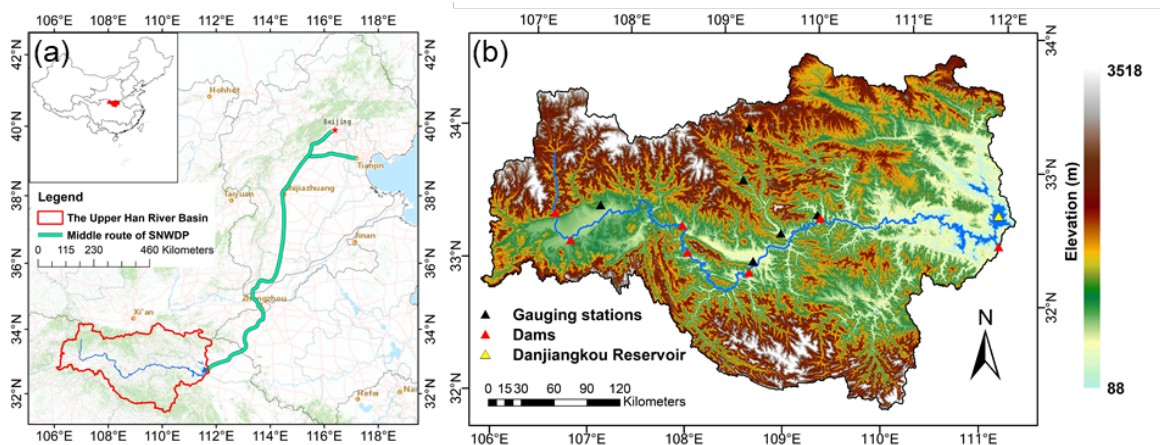

**Figure 1: (a) The location of the UHRB in China and the middle route of the South-North Water Diversion Project (from the Map World); (b) Topography of the UHRB, where the black, red, yellow triangles respectively mark the location of the hydrological gauging stations for model evaluation, hydropower plants (dams), and the Danjiangkou Reservoir.**

The SNWDP is an ambitious plan to alleviate the water shortage in North China, whose water consumption and requirements have increased greatly since the 1980s due to the increasing acceleration of both economic development and population growth (Liu and Zheng, 2002). The SNWDP serves roughly 400 million people, accounting for about one-third of China's population. However, annual available water resources per capita in North China are less than one-quarter of those in South China (Zhang et al., 2020). Water shortages in North China, including in China's capital Beijing, have become a major factor constraining economic and social development. The SNWDP consists of three routes: the eastern, the middle, and the western routes (Liu and Zheng, 2002). The eastern route uses an ancient canal in East China to divert water from the lower Yangtze River, the middle route diverts water from the UHRB via aqueducts, and the western route (currently in the planning stages) would divert water from the upper Yangtze River (Liu and Zheng, 2002).

### 2.2 Data Sources and Processing

The land cover and vegetation index data used in this study were obtained from Moderate Resolution Imaging Spectroradiometer (MODIS) data products (Table 1). Biomes were defined based on land cover data for 2001-2018 from the MODIS annual 500 m land cover product (MCD12Q1 v006), using the International Geosphere-Biosphere Programme (IGBP) classification scheme (Sulla-Menashe et al., 2019). We also obtained 16-day, 250 m monthly normalized difference vegetation index (NDVI) from the MODIS MOD13Q1 v006 product for the same period (Huete et al., 2002). We smoothed the NDVI data with the adaptive Savitzky-Golay filter in the TIMESAT 3.3 software (Jönsson and Eklundh, 2004), then aggregated to a monthly scale with temporal averaging.

The climate data required for the model include precipitation (P), air temperature (T), vapor pressure deficit (VPD, in hPa), and shortwave radiation (SR). These were all obtained from the monthly, ~4 km (1/24 degree) TerraClimate dataset

(Abatzoglou et al., 2018) for 2001 to 2018 (Table 1). We estimated the mean temperature by averaging monthly maximum and minimum temperature.

Soil attributes were derived from the SoilGrids dataset (Table 1), a system for global digital soil mapping using a state-of-the-art machine learning method to map the spatial distribution of soil properties across the globe. SoilGrids prediction models are fitted using over 230,000 soil profile observations from the World Soil Information Service database and a series of environmental covariates. The soil data include sand, silt, and clay content at six standard depth intervals (5cm for each interval) at a spatial resolution of 250 m needed to derive soil model parameters (Table A1). All spatial data were rescaled to 250 m resolution based on the cubic convolution resampling method in ArcGIS 10.5, except the land cover data which was resampled based on the nearest neighbour.

**Table 1: Descriptions of model input data sources and usage by the hybrid ecohydrological model.**

| Dataset | Source | Usage | Spatial and Temporal Resolution | Period |
|---|---|---|---|---|
| Digital elevation model | Shuttle Radar Topography Mission (SRTM) | Drive CCW model & Extract Watersheds | 30m/~ | ~ |
| Land cover | MODIS, MCD12Q1 v006 | Drive CCW model | 500m/Yearly | 2001-2018 |
| NDVI | MODIS, MOD13Q1 v006 | Drive CCW model | 250m/16-day | 2001-2018 |
| Climate* | TerraClimate | Drive CCW and WaSSI | 4km/Monthly | 2001-2018 |
| Soil | SoilGrids from the International Soil Reference and Information Centre | Drive WaSSI | 250m/~ | ~ |
| Measured streamflow | Records of the hydrological gauging stations | Model evaluation | ~/Monthly or yearly | 2009-2015 |

**\* The climate data (http://www.climatologylab.org/terraclimate.html) include precipitation, air temperature, vapor pressure deficit, and shortwave radiation at monthly time steps.**

## 2.3 Model development

We integrated the WaSSI model (Sun et al. 2011; Caldwell et al., 2012) and the CCW model (Zhang et al., 2016), hereafter referred to as the CCW-WaSSI model, to fully take advantage of the strengths of both models and effectively address our research objectives (Figure 2). The 'water-centric' WaSSI model is an integrated ecohydrological model designed for modelling water balance and carbon assimilation at a broad scale. The key components of WaSSI include a parsimonious ET model and a soil water routing model for estimating ET, water yield, and ecosystem productivity (Sun et al., 2011; Caldwell et al., 2012). To account for the biophysical control on the ET processes we replaced the ET model in WaSSI and adopted the carbon-centric ET model in CCW. This modification effectively couples the carbon assimilation (gross primary production,

GPP) and ET processes at a monthly scale (Zhang et al., 2016). As a result, the hybrid model retains mechanistic linkages between carbon and water in the CCW model and watershed soil water routing in WaSSI, and the simplicity and computational efficiency of both models.

CCW has a much simpler model structure than more complex process-based models for ET, e.g., the Penman-Monteith (Penman, 1948; Monteith, 1965), RHESSys (Tague and Band, 2004), and ORCHIDEE (Krinner et al., 2005) models, while maintaining similar accuracy (Zhang et al., 2016, 2019b). Driven by remotely sensed data, CCW first estimates GPP as the product of absorbed photosynthetically active radiation (APAR) and realized light use efficiency ($\varepsilon$) (Figure 2), from which ET is estimated based on underlying water-use efficiency (UWUE) theory (Zhou et al., 2014):

$$GPP = APAR \times \varepsilon = (PAR \times FPAR) \times (\varepsilon_{pot} \times R_s) \times (T_s \times W_s), \tag{1}$$

$$ET = \frac{GPP \times VPD^{0.5}}{UWUE}, \tag{2}$$

where PAR is the photosynthetically active radiation (MJ m$^{-2}$), which is taken as 45% of the total shortwave radiation (Running et al., 2000); FPAR is the fraction of photosynthetically active radiation (PAR) absorbed by plants, determined by NDVI; $\varepsilon_{pot}$ (g·C MJ$^{-1}$) is the biome-specific potential light use efficiency under optimal conditions; $R_s$, $T_s$, and $W_s$ are, respectively, environmental scalars (in the range of [0,1]) related to diffuse radiation, temperature, and moisture stresses to primary production; UWUE represents the biome-specific underlying water use efficiency, derived from global flux tower data (Pastorello et al., 2020), ranging from 4.5 to 8.4 g C/kg H$_2$O. FPAR, $R_s$, $T_s$, and $W_s$ were calculated according to Sims et al. (2005), King et al., (2011), Raich et al., (1991), and Landsberg and Waring, (1997), respectively, as:

$$FPAR = 1.24 \times NDVI - 0.168, \tag{3}$$

$$R_s = 1 - K_1 \times R_a / R_{cs}, \tag{4}$$

$$T_s = \frac{(T - T_{min}) \times (T - T_{max})}{(T - T_{min}) \times (T - T_{max}) - (T - T_{opt})^2}, \tag{5}$$

$$W_s = exp\left(-K_2 \times (VPD - VPD_{min})\right), \tag{6}$$

Where $R_a$ and $R_{cs}$ are respectively actual and clear-sky radiation. The calculation of $R_{cs}$ is based on Raes et al. (2009). $T_{min}$, $T_{max}$, $T_{opt}$ are respectively the biome-specific minimum, maximum and optimal air temperature for photosynthetic activity. $VPD_{min}$ is the biome-specific minimum VPD exceeding which moisture stress starts to take effect. The parameters ($\varepsilon_{pot}$, $T_{min}$, $T_{max}$, $T_{opt}$, $VPD_{min}$, $K_1$ and $K_2$) were calibrated based on global FLUXNET data through a Monte Carlo simulation (Zhang et al., 2016, 2019b).

Using ET estimated from CCW, we estimated WY with WaSSI (Figure 2). The Sacramento Soil Moisture Accounting Model (SAC-SMA) (Burnash, 1995; Burnash et al., 1973) was used to model WY in the WaSSI model, driven by ET, precipitation (P), and soil parameters (Table A1). Soil parameters (Table A1) were generated from multi-layer soil particle-size distribution and depth according to Anderson et al. (2006). The algorithm divides the soil layer into lower and upper zones at different depths and estimates the distribution of moisture—including both tension water components (driven by evapotranspiration and diffusion) and free water components (driven by gravitational forces) in each of these two zones (Figure

2). The model then uses P, soil moisture, and the basin's relative permeability to estimate total water storage and run-off (Figure 2).

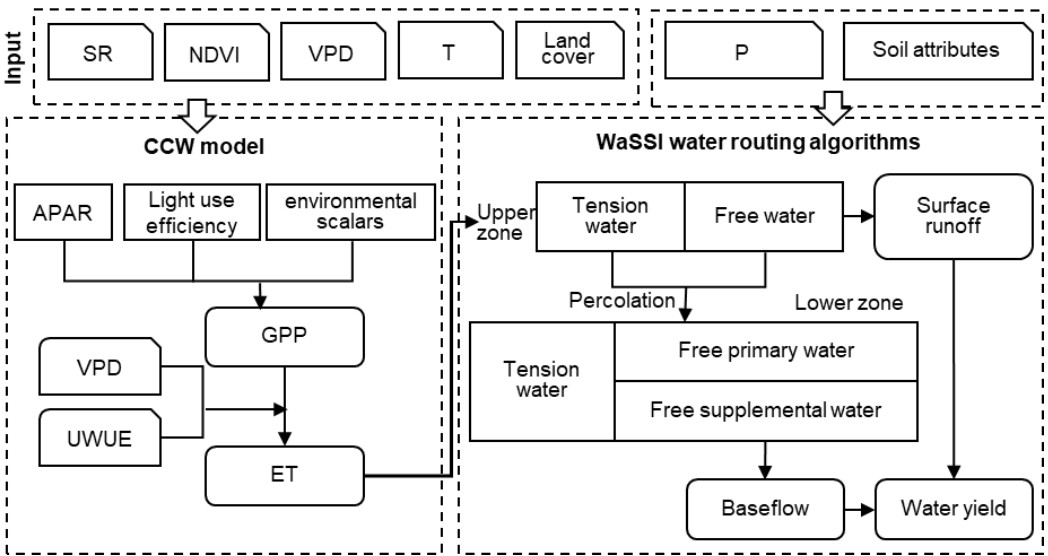

**Figure** 2**: The framework of the CCW-WaSSI hybrid ecohydrology model. The corner-snipped rectangles denote inputs of the model, the rectangles denote state variables, and the rounded rectangles denote fluxes as model outputs. Arrows show the model process directions.**

**2.4 Model Evaluation**

To evaluate the performance of the CCW-WaSSI model, we compared the estimated WY to the measured streamflow of the hydrological gauging stations within the UHRB. We used the six stations within the basin that were relatively free from direct modification (e.g., hydropower plants, reservoirs, dams) (Figure 1b) to evaluate the model both at monthly and annual scale. We also used the annual streamflow records of the Danjiangkou Reservoir to evaluate the modeled WY for the overall UHRB. It should be noted that the streamflow records of the reservoir were under the influences of at least seven major hydropower plants along the Han River mainstream (Figure 1b) and several other hydraulic structures within the UHRB. We evaluated model performance using both the Nash-Sutcliffe efficiency (NSE) and the coefficient of determination ($R^2$). The NSE is a widely used statistic variable for assessing the goodness of fit of hydrologic models, defined as one minus the ratio of the error variance of the modeled time-series WY divided by the variance of the observed WY time-series (McCuen et al., 2006).

**2.5 Model simulations of greening effects on water yield**

To explore the relative contributions of vegetation and climate on WY, we designed three scenario experiments (Table 2). We first simulated the actual variation of WY based on dynamic land cover type, NDVI, and climate from 2001 to 2018 (Scenario S1), thus representing the combined effects from both climate and vegetation. To isolate the effect of vegetation alone on WY, we designed two more simulation scenarios. In Scenario S2, we fixed land cover and NDVI in 2001, while all

climatic variables were allowed to change, thus obtaining climate effects on WY without any vegetation greening. In Scenario S3, we fix land cover and NDVI at 2018 values while allowing the climatic variable to change with time, thus simulating WY after vegetation greening.

We estimated two types of greening effects on WY using a difference-in-difference approach: the dynamic greening effects (S1−S2) and potential greening effects (S3−S2). The dynamic greening effects are the dynamic WY changes caused by vegetation greening alone during 2001-2018. The potential greening effects are the differences in WY between S2 and S3, which is the possible changes in WY from vegetation greening during 2001-2018 if each year had the same vegetation greening condition as 2018. Unlike dynamic greening effects, the potential greening effects can present a range of greening effects with

the variation of climate conditions, which is expected to continue in the future. To investigate trends in WY for each scenario, we used the Mann-Kendall test, a widely used test in hydrological studies for trend and change point detection (Hamed, 2008).

    To quantify changes in hydrological drought risk from vegetation greening, we calculated a hydrological drought index from WY under the three scenarios. Hydrological drought refers to a severe lack of water in the hydrological system, manifesting in abnormally low streamflow in rivers and abnormally low water levels in lakes, reservoirs, and groundwater

(van Loon, 2015). Here, the monthly drought index was calculated as the percentages of monthly WY to the mean WY of the same month during 2001-2018 based on simulated WY (Hayes et al., 2002). We then classified drought intensities for each month based on the magnitude of the drought index. Specifically, months with WY within 10% above or below average were classified as "normal"; months with WY 10%-30% above or below average were classified as "moderate drought/wet"; months with WY 30%-50% above or below average were classified as "severe drought/wet"; and months with WY greater than 50%

above or below average were classified as "extreme drought/wet".

**Table 2: Scenario simulation experimental design to separate the effects of vegetation change from climate effects in the Upper Hanjiang River Basin**

| Scenarios | NDVI and land cover | Climate variables | Purposes |
|---|---|---|---|
| S1 | Dynamic | Dynamic | Estimating actual dynamics of water yield |
| S2 | Fixed in 2001 | Dynamic | Estimating potential water yield without vegetation greening |
| S3 | Fixed in 2018 | Dynamic | Estimating potential water yield after vegetation greening |

## 3 Results

### 3.1 Vegetation and climate changes

The annual mean NDVI over the UHRB showed a significant upward trend with a rate of $0.5\pm0.1$ % $yr^{-1}$ at the 95% confidence level (Figure 3a). Spatially, 97.4% of the area had increasing trends, 94.0% of which were statistically significant ($p < 0.05$). The few areas with decreasing trends were distributed around the cities and the Danjiangkou Reservoir. In total,

17.9% of the land area in the basin experienced a change in land cover between 2001 and 2018. Forest cover increased from 40.9% (38,753.6 km$^2$) in 2001 to 50.7% (48,012.6 km$^2$) in 2018 (Figure 3b), and 98.8% of the new increased forests were converted from shrubland. Shrubland showed the largest decrease by 18.3%, from 51.8% (49,093.4 km$^2$) in 2001 to 42.3% (40,025.5 km$^2$) in 2018 (Figure 3b). The area of cropland decreased by 9.9% from 5,175.1 km$^2$ (5.5%) in 2001 to 4660.8 km$^2$ (4.9%) in 2018. The area covered by open water more than doubled, from 313.1 km$^2$ in 2001 to 746.6 km$^2$ in 2018, likely caused by increasing water levels in the Danjiangkou Reservoir for providing water to the SNWDP.

Due to high interannual variability and the relatively short study period (less than 20 years), none of the four climatic variables (i.e., P, T, VPD, SR) showed statistically significant trends at the annual scale (Figure A1 in Appendix).

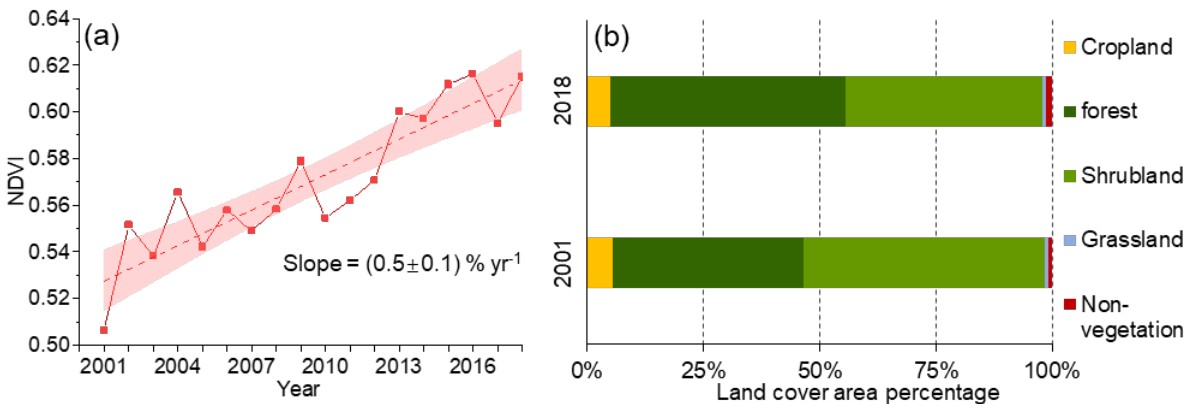

**Figure 3: (a) The temporal variation of annual mean NDVI in the UHRB during 2001-2018 and its 95% confidence interval (red shade); (b) The composition of the main land cover types in the UHRB in 2001 and 2018.**

## 3.2 Model evaluation

The CCW-WaSSI model performed well in estimating monthly and annual WY at six hydrological gauging stations in the UHRB. The measured and simulated WY are not only strongly correlated with each other but also have nearly identical trends (-0.3±1.0 km$^3$ yr$^{-1}$ at the 95% confidence level) (Figure 4a) at the entire watershed scale. The CCW-WaSSI model captured 90% of annual WY variation (R$^2$ = 0.9), with an NSE of 0.8 and root mean squared error (RMSE) of 3.9 km$^3$ yr$^{-1}$ (Figure 4a) for the whole UHRB. At the sub-watershed scale based on the six hydrological gauging stations, the model captured 80% of the annual WY variation (R$^2$ = 0.8), with an NSE of 0.9, RMSE of 96.3 mm yr$^{-1}$ (Figure 4b). Monthly-scale NSE across the six stations ranged 0.5-0.8 (Figure 4c), and the model captured 50%-80% of the monthly variation (R$^2$ = 0.5-0.8) of WY with RMSE of 15.9-30.7 mm per month (Figure 4c). Averaged across all gauges, monthly NSE was 0.6, R$^2$ was 0.7, and RMSE was 21.2 mm per month.

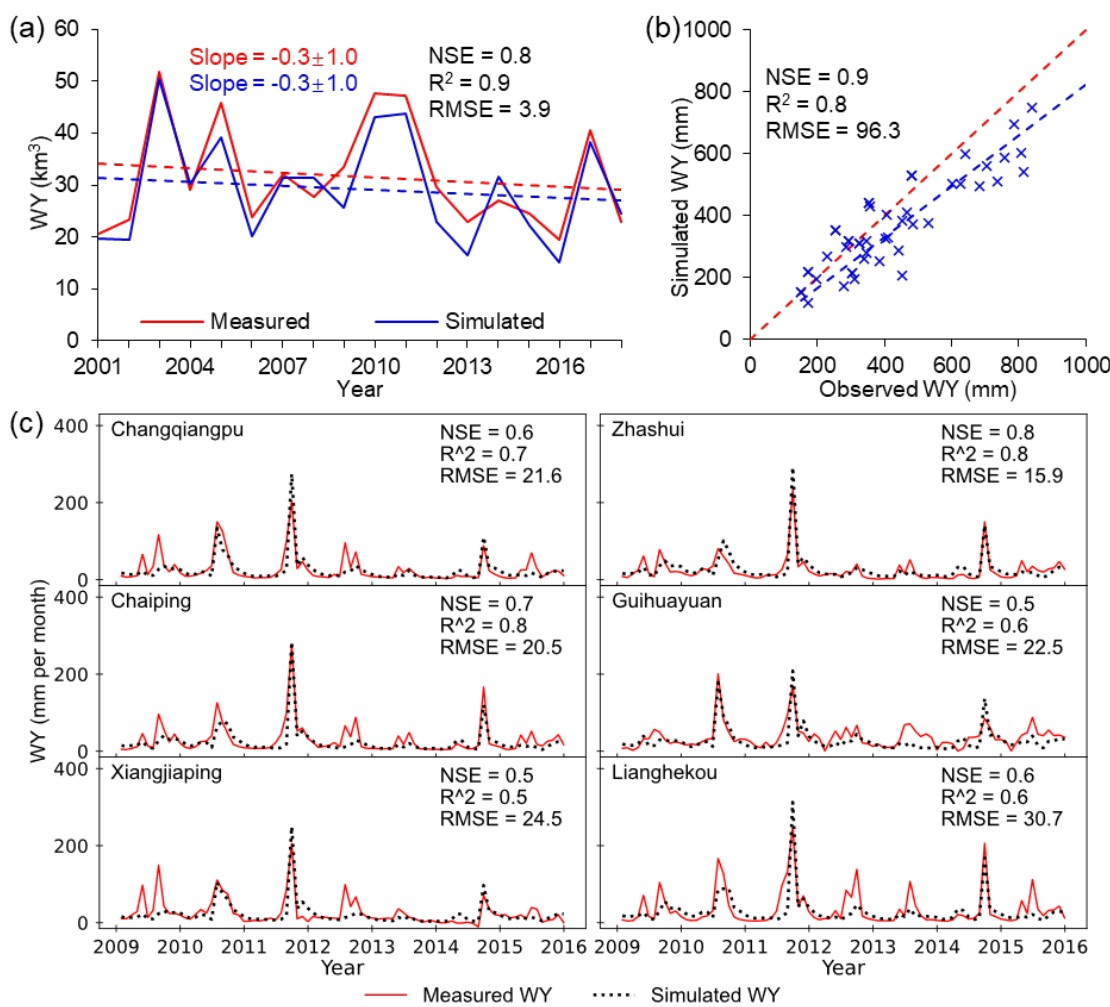

**Figure 4: (a) Time series of the simulated water yield (WY) and measured streamflow to the Danjiangkou Reservoir at the annual scale, and their slopes at 95% confidence intervals; (b) Comparison between simulated and observed annual WY for six hydrological gauging stations within the UHRB, shown in Figure 1; (c) Time series of the simulated and measured WY at the monthly scale for six hydrological gauging stations.**

### 3.3 Changes in water yield (WY)

The UHRB had an annual WY of 160-533 mm yr$^{-1}$ with a mean of 309 mm yr$^{-1}$ during 2001-2018 under Scenario 1 (i.e., all factored considered) (S1), (Figure 5a, 5b), peaking in late summer to early autumn and accounting for 34% of the mean annual P (Figure 5d). The overall WY in the UHRB slightly decreased at a rate of -2.9 ±10.0 mm yr$^{-1}$ over the study period at the 95%

confidence level. Decreasing trends of WY occurred over 74% of the UHRB, though only 9% of the basin had a trend at the

confidence level of 90% (p < 0.10; see Appendix Figure A2). The model experiments (Scenario S1 and S2) revealed that vegetation greening had a significant negative effect on annual WY with a rate of -3.2 ±1.0 mm yr$^{-1}$ at the 95% confidence level during 2001-2018 (Figure 5c). Spatially, greening induced a WY decrease in 90% of the UHRB, while WY increases due to greening were mainly in high elevation areas (above 3000 m). The WY decrease from greening had strong negative correlations over space with elevation (R=-0.7), and positive correlation with average annual temperature (R=0.7) and VPD (R=0.8). The effects of climate on WY varied substantially from year-to-year, with a standard deviation (STD) of 101.9 mm, but had no significant trend (0.4±10.0 mm yr$^{-1}$ at the 95% confidence level; Figure 5c). These climate effects were also the main driver of overall annual WY variation (STD = 108.6).

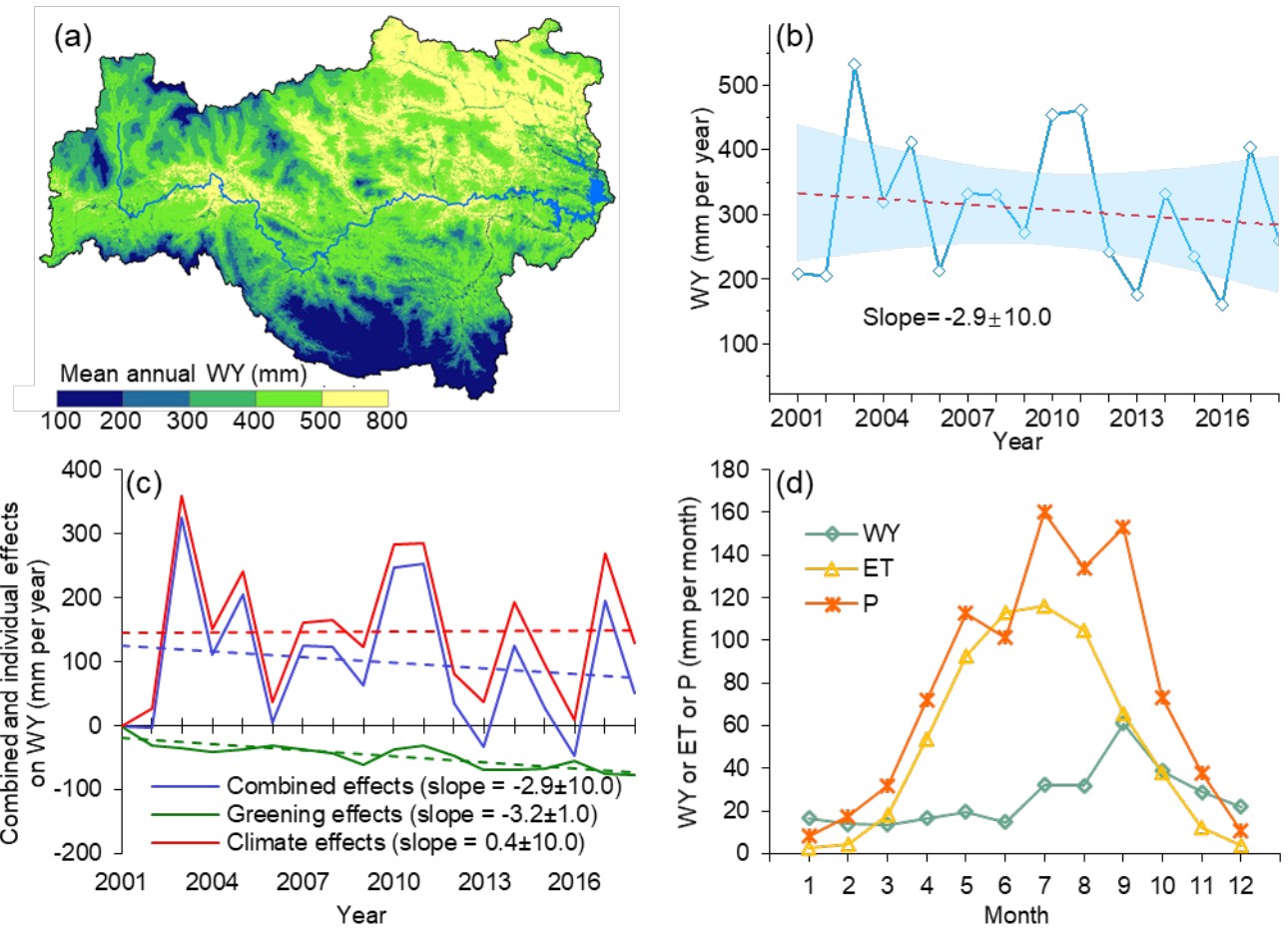

Figure 5: (a) The spatial distribution of modeled mean annual water yield (WY) during 2001-2018 in the Upper Han River Basin; (b) The temporal variation of modeled annual WY (Scenario S1) during 2001-2018, and its 95% confidence interval (blue shade) ; (c) The combined and individual effects of vegetation and climate on WY during 2001-2018, and their slopes at 95% confidence intervals; (d) Average monthly evapotranspiration (ET), WY, and precipitation (P) during 2001-2018.

In addition to reducing total WY, vegetation greening also significantly reduced the ratio of annual WY to P (Figure 6a). After 2003, the ratio decreased with a significant trend of 0.01±0.008 yr$^{-1}$ at the 95% confidence level (Figure 6a). The annual

WY/P ratio also had a significant negative correlation with NDVI, with a correlation coefficient of −0.7 (p < 0.001) (Figure 6b).

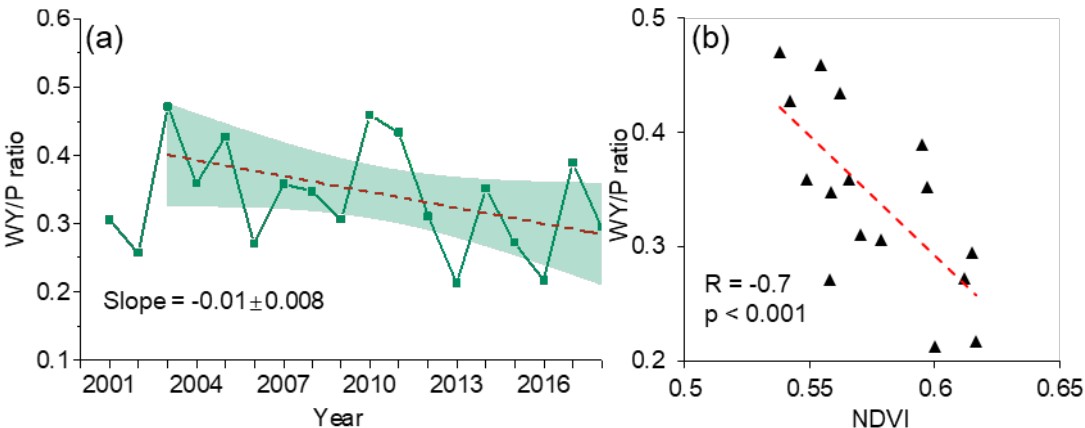

Figure 6: (a) The temporal variation of annual WY/P (water yield/precipitation) ratio, the red dotted line denotes the trend for the period 2003-2018, the green shade is the 95% confidence interval; (b) The scatter plot between WY/P ratio and NDVI during 2003-2018.

### 3.4 Potential greening effects on WY in different climate conditions

The greening trend from 2001 to 2018 significantly reduced WY, but the same greening trend could exert different effects on WY depending on climate conditions (Figure 7). By comparing the WY without greening (Scenario S2) and after greening (Scenario S3) but with dynamic climate variability, we found that the vegetation greening could potentially induce a reduction of 77.1 mm in annual WY on average, accounting for 25% of the mean annual WY (308.5 mm) during 2001-2018. The 2018 vegetation greening condition could have caused the largest decrease in WY by 99.3 mm in 2003 and the least by 43.9 mm in 2001 (Figure 7). The relative changes in WY derived from S2 and S3 during 2001-2018 ranged from 14% (2011) to 31% (2002) (Figure 7).

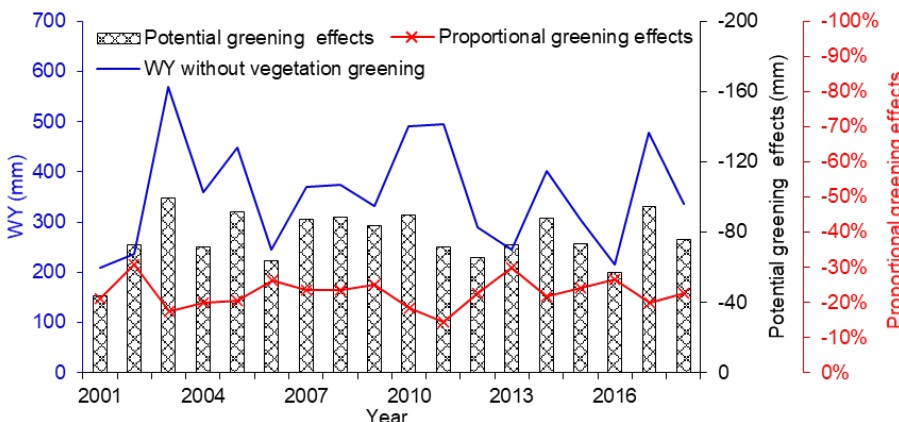

**Figure 7: The annual water yield (WY) without vegetation greening and potential effects of vegetation greening on WY from 2001 to 2018 derived from the difference between S2 and S3 under different climate condition and their proportion to WY without greening.**

The differences in potential vegetation greening effects on WY could result from different climate conditions in those years. In order to understand the relationship between the potential greening effects on WY and climate, we explored the correlation between greening-induced ET and WY changes and climate variables (Figure 8). The three climate variables (SR, T, VPD) have strong positive correlations (R > 0.6) with potential absolute ET changes from greening (Figure 8), indicating that vegetation greening would increase ET more in a warmer and drier climate. In contrast, the three climate variables are not as

strongly correlated with potential absolute WY changes from greening (Figure 8). The nature of the relationship reversed for potential relative WY changes from greening compared with those of the absolute WY changes. The greening effects on proportional change in WY were positively correlated with T (R = 0.7, p = 0.02), SR (R = 0.5, p = 0.00), and VPD (R = 0.4, p = 0.06) (Figure 8), and negatively correlated with P (R = -0.7, p = 0.00), indicating that vegetation greening could cause more proportional WY decrease in dry years. Regression analyses also revealed that the proportional WY changes from greening

would increase one percentage point per 0.1°C increase in T, or 0.3 hPa increase in VPD.

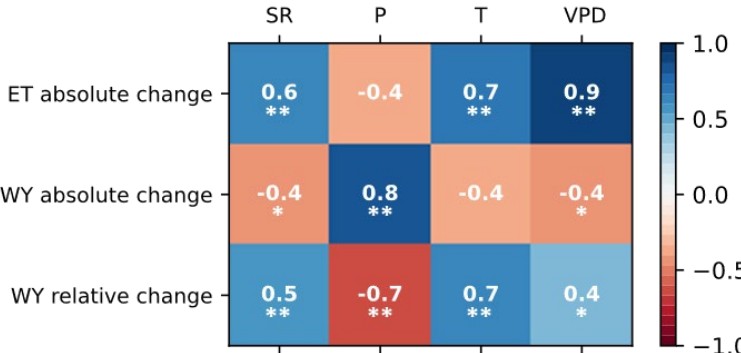

**Figure 8: Correlation coefficients between potential greening effects on evapotranspiration (ET) and water yield (WY) and climate variables (shortwave radiation, SR; precipitation, P; temperature, T; and vapor pressure deficit, VPD). The absolute changes refer to total magnitude change in ET or WY from vegetation greening, while the relative changes denote the proportional change in WY after greening relative to WY without any greening. The numbers in boxes are the corresponding correlation coefficients. Double asterisks denote p<0.05; single asterisks denote p<0.10.**

### 3.5 Changes in hydrological drought risk from vegetation greening

Hydrological drought risks increased due to vegetation greening. There were 109 drought months during 2001-2018 based on scenario S1 (dynamic climate and vegetation), but there were only 87 drought months of the same intensity in the scenario without greening (S2) (Figure 9). In contrast, the number of hydrological drought months increased to 132 in the scenario with 2018 greenness (S3) (Figure 9). The risk of extreme hydrological drought more than doubled between the scenarios without greening (17 total months) and after greening (42 total months), indicating that vegetation greening during 2001-2018 could not only increase the frequency of hydrological drought relative to historical norms but could also amplify the intensity of those hydrological droughts.

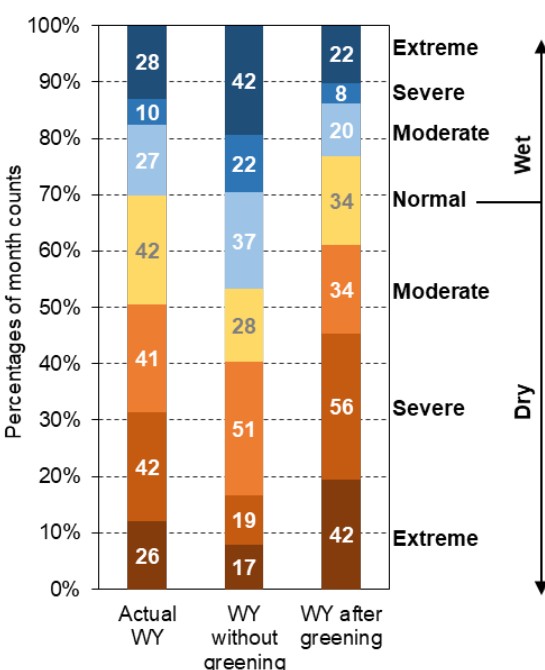

**Figure 9: The percentages of counts of months in the different hydrological drought intensities for actual water yield (WY) (S1), WY without greening (S2), WY after greening (S3) (numbers in color bars are the corresponding counts of months).**

Given that the monsoon period (July-November) contributes about 70% of annual WY in the UHRB, changes in hydrological drought risk during the monsoon period is more critical to water supply capacity to the water diversion project than other months. Nearly two-thirds (63%) of months experiencing severe or extreme hydrological drought for scenario S1

were during the monsoon periods (July-November) of 2001-2018. Vegetation greening could potentially cause a reduction in WY during the monsoon period of 38.2 mm on average, accounting for 18% of the mean WY without greening during the monsoon period. The six driest monsoon periods were in severe or extreme hydrological drought (less than 70% of mean WY) during 2001-2018, during which vegetation greening could potentially cause a WY decrease of 23.2 mm, accounting for 23% of the WY without greening.

## 4 Discussion

### 4.1 Reduced water yield and drought amplification from greening

Paired catchment and modelling studies provided evidence that vegetation greening and/or afforestation would increase ET and decrease WY (Bosch and Hewlett, 1982; Farley et al., 2005; Liu et al., 2008). However, the extent of the effects of vegetation greening varies in different climate conditions (Bai et al., 2020; Ingwersen, 1985). Typically, WY is more sensitive to vegetation greening in water-limited regions than in energy-limited regions (Feng et al., 2016). For example, in the Loess Plateau, an arid/semi-arid area of China, the ratio of annual WY to P decreased from 8% before afforestation (1980-1999) to 5% as vegetation increased during 2000-2010 (Feng et al., 2016). In contrast, the Poyang Lake basin, a subtropical basin in southeast China with an annual P of around 1,800 mm, experienced greening as well, but its effects on annual WY were limited (Guo et al., 2008; Wang et al., 2018). Unlike the previous two examples, vegetation greening in the UHRB induced a substantial decrease in WY (scenario S2), but WY under the combined effects (scenario S1) did not have a statistically significant trend as a result of the large interannual variation of the climate.

The UHRB is located on the southern side of the Qinling Mountains, which marks the northern edge of the subtropical monsoon and the dividing line between the subtropical and temperate zones in China (Figure 1a). The P in the UHRB is less than that of other subtropical basins (e.g., Poyang Lake basin), but ET of the UHRB is comparable to those subtropical basins due to the abundance of vegetation. The climate regime and vegetation cover make the UHRB more likely to suffer hydrological drought, and the water cycle can be largely influenced by vegetation dynamics. While the WY of the Poyang Lake basin would only decrease approximately 3% even under an extreme greening scenario (Guo et al., 2008; Tang et al., 2018), the effects of vegetation greening on the water cycle in the UHRB are much larger than this, with a decrease in WY up to 25% for comparable levels of vegetation greening (Figure 5c). However, as the latest round afforestation efforts are about to conclude, implementation of afforestation projects will likely soon slowdown in China, and combined with potential limitations from future water stress (Feng et al., 2021), the current greening trend may not continue or even slowdown in the UHRB in the future.

Although vegetation greening had negative effects on WY at the regional scale as we found in the UHRB, some previous studies indicated that vegetation greening could increase WY at larger spatial scales (Ellison et al., 2012; Makarieva et al., 2007; Spracklen et al., 2012). Enhanced ET from vegetation could moisten the atmosphere, thus drive P to increase (Ellison

et al., 2012). On the other hand, Makarieva et al., (2007) argued that increases in ET could drive the transport of water vapor across continental space via changing atmospheric pressure dynamics. Therefore, vegetation greening in the UHRB may also potentially increase P via increasing atmospheric vapor. However, the amount of recycled P is strongly dependent on the watershed area, with larger geographical expanses having greater potential for recycling (Ellison et al., 2012). At the regional scale, 87% of the atmosphere moisture through enhanced ET in nine large basins around the world are not likely to recycle back as P, but transport to other regions (Ellison et al., 2012). The proportion of recycled P in the UHRB would be lower due to its smaller extent. Tuinenburg et al., (2020) found that moisture from ocean contributes approximately 67% of P in the Yangtze River Basin (where the UHRB located). In addition, the atmosphere moisture through enhanced ET in the upwind areas was possible to transport to the downwind, thus increase P (Ellison et al., 2017). The upwind areas of the UHRB, Southeast China, also experienced vegetation greening simultaneously (Chen et al., 2019), which may lead to P increase in the UHRB. A modelling study found that vegetation greening only induced an P increase of 1.5% per decade in the Yangtze River Basin from 1982 to 2011 (Li et al., 2018). Given that no particular trend in annual P in the UHRB was observed (Figure A1b), the effects of local and upwind greening on P may be limited during the study period. However, such complex feedbacks among vegetation, evaporation and precipitation are worth investigating in the future.

Aside from greening, temperature and VPD are also critical factors in the interaction between vegetation and the water cycle. Drought risk has increased globally in recent decades and will likely continue to increase in the future as a result of anthropogenic climate change (Cook et al., 2020; Huang et al., 2016; Lesk et al., 2016; Williams et al., 2020). An increase in temperature increases saturation vapor pressure of the atmosphere, which in turn increases global VPD (Yuan et al., 2019; Zhang et al., 2019a), especially when combined with a decrease in oceanic evaporation (Trenberth et al., 2007), which contributes approximately 85% of atmospheric water vapor. Here, we show that the same greening trend could cause a greater decline in WY under higher temperature and VPD conditions.

### 4.2 Trade-offs among ecosystem goods and services induced by greening

One of the initial intentions of afforestation and vegetation restoration projects in the UHRB was reducing soil erosion and sedimentation, thus improve water quality in the downstream. The projects had achieved these goals to some extent according to government reports and related studies (Qi et al., 2019; Li et al., 2008). Vegetation recovery in the UHRB reduced annual sediment by $1.7 \times 10^7$ metric tons from 2000 to 2010, representing 13% of sediment loading in 2000 (Qi et al., 2019). An in-situ sample analysis found that the watersheds in the UHRB with higher vegetation cover had relatively better water quality in terms of turbidity, suspended particulate matter, and dissolved solids (Li et al., 2008). Zhang et al. (2014) found that vegetation greening also had additional benefits through enhancing vegetation productivity and potentially increasing carbon sequestration in the Yangtze River Basin, in which UHRB as a sub-basin is a major contributor.

On the other hand, as indicated by our study, the afforestation and vegetation greening could decrease WY in the UHRB through enhancing ET, and thus reduce the water supply capacity to SNWDP. Although the capacity of the water supply to

the SNWDP is directly linked with the holding capacity of the source reservoir (i.e., Danjiangkou Reservoir; Figure 1b), it is influenced by the net water yield produced by UHRB, which can be approximately estimated as WY of UHRB minus the downstream water demand. The downstream water use demands include household, agriculture, and industry, as well as a basic flow maintaining rate for shipping and pollutant dilution (Li et al., 2017; Liu et al., 2003), likely ranging from 12.2 to 18.5 $km^3$ $yr^{-1}$ (Hu and Guo, 2006; Li et al., 2017; Liu et al., 2003; Xu and Chang, 2009). In the future, as our modelling

experiments show (i.e., scenario S3), the water supply capacity to the SNWDP could be 14 $km^3$ $yr^{-1}$ on average even if vegetation greenness does not continue to increase beyond the 2018 level and the climate regime remains stable by assuming the lower bound of downstream water demand as 12.2 $km^3$ $yr^{-1}$. In contrast, the water supply capacity would be about 21 $km^3$ $yr^{-1}$ if the UHRB had pre-2001 vegetation conditions (Scenario S2), indicating that vegetation greening during 2001-2018 could decrease the water supply capacity of UHRB to the SNWDP by 7.3 $km^3$ $yr^{-1}$, accounting for 77% of the planned annual

water diversion target of the Phase I project (9.5 $km^3$ $yr^{-1}$).

The capacity of water supply to the SNWDP in the source Danjiangkou Reservoir was highly influenced by the seasonal water yield under extreme climate conditions such as drought in the UHRB. Vegetation greening could significantly exacerbate the risk, duration, and intensity of hydrological drought through enhancing ET, although the possible negative feedback of drought on vegetation could slow down such effects. Consequently, the water storage of the reservoir would rapidly decrease

as hydrological drought progresses, thus directly lowering the water diversion potential to the SNWDP, although the active reservoir management may reduce the water shortage risk for SNWDP. In such scenarios, SNWDP would have to face a dilemma to balance the trade-off of forest ecosystem services induced by the ongoing afforestation and vegetation greening: local WY was significantly reduced despite an improvement in water quality (Li et al., 2008) and carbon uptake (Zhang et al., 2014).

## 4.3 Implications for water diversion projects

Compared to other water diversion projects (WDPs) around the world, the middle route of the SNWDP diverts a much larger proportion of water yield from the source basin (UHRB) (Shumilova et al., 2018). About one-third to half of the total annual WY from UHRB is diverted by the SNWDP. In contrast, 78% of the water diversions in the US in 1973-1982 extracted <1% of annual streamflow from the source basins (Emanuel et al., 2015). Consequently, the middle route of the SNWDP has a large

influence on the natural water balance in the water source basin, and the water diversion capacity is more vulnerable to hydrological drought events. In addition, more than 12 million people live downstream of the UHRB, and 12,000 $km^2$ of farmlands and many large-scale industries all rely on water from the UHRB. Currently, the perennial mean water available for diversion is 12-14 $km^3$ $yr^{-1}$, but only about 6 $km^3$ $yr^{-1}$ in dry years (Wang and Yang, 2005). Water supply under severe droughts could be much lower (Barnett et al., 2015), and increasing frequency or severity of drought events will challenge the feasibility

of the project (Liu et al., 2015). The inflow of the Danjiangkou Reservoir showed a sharp decrease from 41.0 $km^3$ during 1951-1989 to 31.6 $km^3$ $yr^{-1}$ during 1990-2006, largely attributed to the reduction of precipitation (Liu et al., 2012). According to

official statistics of the SNWDP (middle route), the capacity of the water supply of the UHRB may not be able to meet expectations, partly due to the insufficient WY in the UHRB during the operational period relative to long-term levels (Zhang et al., 2020). To stabilize the water supply of the middle route of the SNWDP, the Phrase II project will construct an additional
415 canal to divert water from the mainstream Yangtze River to the Danjiangkou Reservoir.

Watershed management to reduce soil erosion and sedimentation is essential in maintaining headwater watersheds for clean water supply in WDPs (Li et al., 2008). Our study suggests that, with recovery of forest vegetation and associated 'greening', water yield could be significantly reduced, especially during drought periods. Therefore, navigating the trade-off between water quality improvement and water yield and supply should be an important consideration for current and future WDPs. To
420 achieve this, we suggest that local rather than fast growing exotic tree species should be used to minimize water use while still serving soil erosion control and ecological restoration purposes (Sasaki et al., 2008; Cao et al., 2011). Soils in natural forests may have more organic matter and higher surface infiltration rates compared to plantation forests in southern China (Yang et al., 2019), and hence generate less sediment (Piégay et al., 2004). Forest management measures, such as stand thinning to reduce water use and fire risk, should also be considered as part of integrated watershed management strategy.

Managing water stress needs to consider both supply and demand. From the perspective of water demand in North China, water transfer from the south supplied more than 70% of the domestic water use in the project-served cities in 2019. Moreover, the probability of concurrent drought events between the UHRB and North China is highly likely to increase in the next 30 years (Liu et al., 2015). Consequently, any fluctuation of water yield or water quality problem in the UHRB may influence the water supply of the serviced cities with a huge population. Therefore, it is risky for cities in North China to become over-
dependent on water diversion. Comprehensive water management policies are needed to sustain the water supply from the UHRB, thus maximumly meet the water demand in North China.

**4.4 Limitations**

Several limitations are worthy of being mentioned. First, we designed three modelling experiments to separate the effects of vegetation greening from those of climate change on WY. Using a new modelling tool, we found that vegetation greening
significantly affected water yield from 2001-2018. However, it is impossible to completely decouple the effects of climate change and vegetation greening on hydrological dynamics due to the tight biophysical interactions and feedbacks between vegetation and climate. The observed vegetation change, as indicated by NDVI, represents the combined effects of changes in climate, human activity (i.e., reforestation, irrigation), and natural vegetation (re)growth. However, because P, T, VPD and radiation had relatively small and insignificant trends during the study period (figure A1), the effects of climate on the
vegetation change (NDVI trend and land cover change) were likely relatively minor compared to direct human modification of the landscape and vegetation growth. Second, as stated before, vegetation greening in the local and upwind area may potentially increase P downwind via increased atmospheric vapor. This water vapor cycling likely offsets some negative local vegetation effects of increased ET on WY. The climate data used as model inputs might implicitly include a certain level of

the feedback effects from vegetation greening, but those effects could not be explicitly disentangled in our analysis. Thus, our results represent an attempt to estimate the direct, first-order net effects of climate and vegetation greening on WY. Third, the potential effects of $CO_2$ fertilization on WY were not explicitly included in this study, although previous work has shown that forest water-use efficiency only increased by about 15-20% over the entire 20[th] century (Frank et al., 2015) and the fertilization effect is weakening (Wang et al., 2020). Lastly, the future effects of vegetation greening on hydrological dynamics and thus water supply for SNWDP depend on the projected warming and drying climate, the continuance of vegetation greening, and the complex feedbacks among climate, soil and vegetation, which deserve long-term monitoring and deeper investigations in the future.

## 5 Conclusions

Using a coupled watershed ecohydrological model, we found that vegetation greening significantly decreased annual water yield at an order of $3.2\pm1.0$ mm $yr^{-1}$ ($p < 0.05$) in the Upper Han River Basin during 2001-2018. Vegetation greening exacerbated hydrological drought risk and reduced about a quarter of water yield on average during hydrological drought periods. This decline in water yield due to vegetation greening has the potential to reduce the capacity of water supply to the South to North Water Diversion Projects, a key project serving water for 400 million people in North China. Despite the enormous ecosystem goods and services provided by forest (e.g., carbon sequestration, water quality improvement, and regulation of air temperature and moisture), our study suggests that navigating the trade-off of water supply with these benefits in source watersheds is an important consideration in large water diversion projects. Integrated watershed management (e.g., forest management practices for reducing water use in headwaters) becomes increasingly essential in vegetation 'greening up' regions, especially in the context of a changing climate and increasing water demand for human use.

*Author contributions.* J. Z. designed the study, developed the model code, did the simulation experiments, and wrote the first draft of the paper; Y. Z. and G. S. and C. S. designed the research and edited the manuscript; M. P. D. and L. H provided feedback on results and edited the manuscript; J. L. offered suggestions on research design; N. L. provided WaSSI model codes for data processing; K. Z. and Q. Z. provided the streamflow data for model evaluation and provided feedback on the modelling results.

*Competing interests.* The authors declare that they have no conflict of interest.

*Code and data availability.* Data sets used for driving model were obtained from different sources described in the Table 1. The monthly streamflow records of six hydrological gauging stations for model evaluation were derived from Yearbook of the Han River Hydrology. The inflow records of Danjiangkou Reservoir were derived from http://113.57.190.228:8001/web/Report/BigMSKReport#. All the data related to our results in this study can be found at

https://osf.io/f5bgk/, except the monthly streamflow records for six hydrological gauging stations are available upon reasonable requests.

*Acknowledgements.* This study was partially supported by the National Natural Science Foundation of China (grants 42061144004) and the overseas' study scholarship offered by the China University of Geosciences Wuhan, China. The majority of the research was conducted in the Remote Sensing and Ecological Modeling group in the Department of Geography at the University of North Carolina at Chapel Hill, USA. Partial support for this study also comes from the Southern Research Station, USDA Forest Service.

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

## Appendix A:

**Table A1: Soil parameters for driving CCW-WaSSI model**

| Parameter types | Parameters | Abbreviations |
| --- | --- | --- |
| Soil Storage | Upper Layer Tension Water Capacity | UZTWM |
| Parameters | Upper Layer Free Water Capacity | UZFWM |
| | Lower Layer Tension Water Capacity | LZTWM |
| | Lower Layer Supplemental Free Water Capacity | LZFSM |
| | Lower Layer Primary Free Water Capacity | LZFPM |
| Baseflow | Depletion Rate from LZFPM | LZPK |
| Discharge Rate | Depletion Rate from LZFSM | LZSK |
| Parameters | Interflow Depletion Rate from UZFWM | UZK |
| Upper to Lower | Percolation fraction direct to LZFW | PFREE |
| Layer Percolation | Percolation Curve Shape | REXP |
| Parameters | Maximum/Minimum Percolation Rate Ratio | ZPERC |

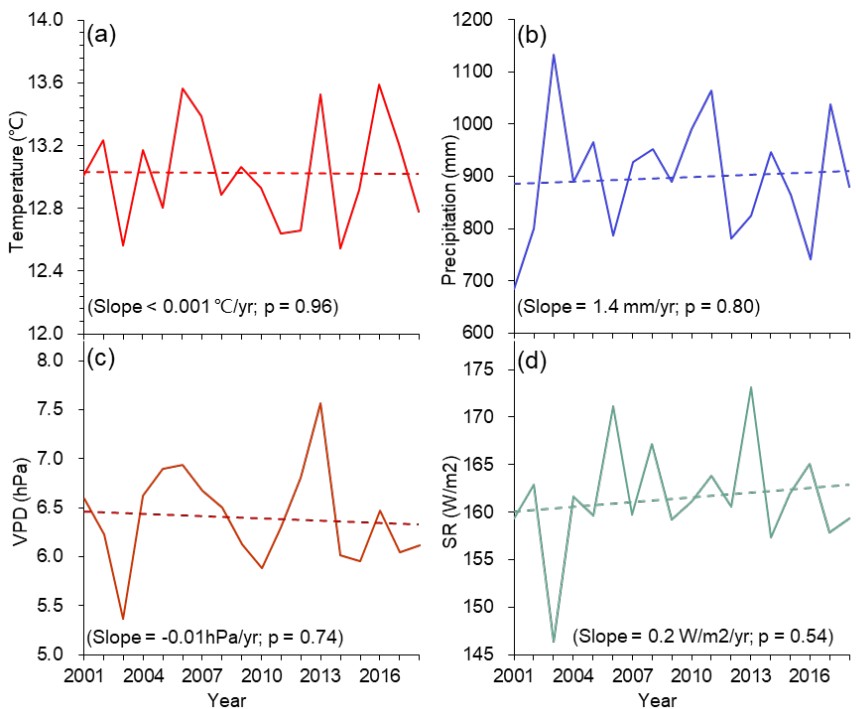

**Figure A1. Inter-annual variations of averaged air temperature (a), total precipitation (b), vapor pressure deficit (VPD) (c) and shortwave radiation (SR) (d) in the Upper Han River Basin from 2001 to 2018.**

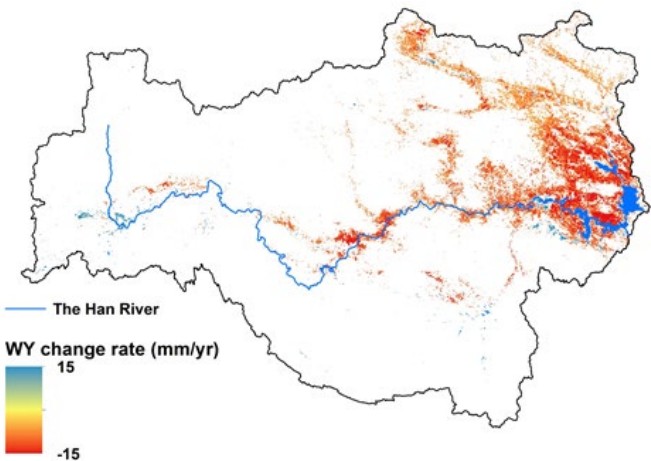

**Figure A2. The linear trend of annual water yield (WY) at the confidence level of 90% (p < 0.10).**