# Peer review of "Vegetation Greening Weakened the Capacity of Water Supply to China's South-North Water Diversion Project"

_Hydrology and Earth System Sciences, 2021_

## Author Comment (AC1)

**To Reviewer #1,**

Comments:

In this manuscript the authors aim to isolate the effects of vegetation greening on water yield in a large basin that serves as donor basin for a major water diversion project. To do so the authors designed a modelled-based scenario analysis. The overall finding of their analyses suggests that greening has the potential to considerably reduce basin water yield and thus supply for the water diversion project. While the experimental set-up is systematic and, in principle, logical and the manuscript is well written, I nevertheless have a number of serious concerns that need to be addressed and resolved in detail before this manuscript could be considered for publication.

Response: We appreciate the reviewer's insightful and constructive comments, which helped us a lot in improving this manuscript. Please find our specific responses below (in blue).

Comments:

(1) The experiment is designed, the results are interpreted and, as a consequence the manuscript is framed from a purely engineering perspective with focus on water yield available for the diversion project. As a consequence, the non-explicit message that is delivered between the lines here is the following: to secure water supply for the diversion project greening needs to be reduced. Or in other, more explicit words: stop afforestation - chop down the forest! I am not sure that this can and should be the message to be conveyed here for the simple reason that this analysis is not comprehensive enough to draw this conclusion.

Response:

This is a valuable and important point! Indeed, as suggested by the reviewer, we did not provide a balanced message about the positive ecosystem goods and services that forests provide, but we mainly focused on the water supply. It was not our intent to have the readers take "stop afforestation-chop down the forest" as the message between the lines from this work. Rather, our message is that despite the myriad ecosystem goods and services forests provide, there can be tradeoffs to these services. We demonstrated that rapid increase in forest cover (i.e., greening up) can significantly reduce freshwater supply, which is itself an essential societal good.

In the revised version of the manuscript, we highlighted positive ecosystem services forests provide, including significantly reducing sediment in the streamflow, improving water quality, and carbon sequestration to mitigate global warming. At the same time, it is important for policy makers to understand that there can be unintended consequences from afforestation and that programs focused on reforestation/afforestation should be

planned accordingly, especially in the context of a warming climate. These tradeoffs in reforestation/afforestation programs have been seen around the world, especially in water shortage regions. For example, a recent report in the Kathmandu Post (https://kathmandupost.com/editorial/2019/12/23/an-unwise-decision-to-plant-conifers-is-parching-the-land) shows that the rapid plantation of pine forests in Nepal jeopardized people's livelihoods due to excessive use of water by the trees. The water yield in most part of the study watershed is sensitive to vegetation change where precipitation is less than 900 mm/yr.

What we intended to stress in this study is the balance between environmental improvement and water resource availability for water diversion projects. Our study was from the perspective of water yield alone. Therefore, what we demonstrated here is a risk of water supply reduction due to afforestation. We advocate for a comprehensive watershed management strategy to deal with water use by forests under a warming future.

Specifically, we made following revisions in the Discussion and Conclusions sections to deliver a more balanced message, highlighting both the goods and services provided by forests and the potential tradeoffs and unintended consequences of afforestation:

a) In "4.1 Reduced water yield and drought amplification from greening", we discussed the greening effects more conservatively. Specifically, we stressed that "However, as the latest round afforestation efforts are about to conclude, implementation of afforestation projects will likely soon slowdown in China, and combined with potential limitations from future water stress, the greening trend will not likely continue at their same levels in the UHRB." And the discussion about the future leaf area increases was removed.

b) The "4.2 Reduced capacity of water supply to SNWDP from greening" was revised as "Trade-offs among ecosystem goods and services induced by greening" and its content was reframed. Specifically, to emphasize the "trade-off" instead of water availability alone, we substantially discussed water quality improvement, as well as other ecosystem goods and services due to greening in the first paragraph of 4.2. Then we shortened the discussion on how greening reduced water availability and moved the discussion about water supply capacity to the water diversion project to section 4.3.

c) In "4.3 Implications for water diversion projects", we provided more discussion on watershed and forest management. Specifically, we moved the discussion about water supply capacity to the water diversion project from 4.2 to combine with the initial first paragraph of 4.3 to discuss the water supply concerns of water diversion projects. Then we offered more specific recommendations for

alleviating effects of greening on water supply. For example, we suggest that using natural regeneration with local tree species rather than artificial plantations to control erosion and conduct ecological restoration. Forest management such as stand thinning to reduce water use and fire risk and increase resilience of the ecosystem should be also considered as part of the integrated watershed management.

d) We added "4.4 Limitations" to discuss the limitations of this study in terms of experiments and external impact factors. We first discussed the possible influences of the interaction between vegetation and climate due to limitations of the experimental design. We then substantially discussed the uncertainty from possible moisture recycling. The enhancement of local ET had the potential to increase P. However, the P in the UHRB is primarily controlled by the Monsoon, which comes from the Pacific Ocean in southeast. The significant and widespread greening was also observed in the southeast China and evaporated a larger amount of moisture which might be brought to the UHRB and potentially induce a P increase. If this happens, such precipitation should already been capture by the precipitation data we used. In the last part of 4.4, we discussed the possible effects of increasing atmospheric $CO_2$ on water cycle.

Accordingly, we revised the conclusion to stress the "trade-off" and suggest that the "improved watershed management (e.g., forest management and reducing water use) is needed to maximum the ecosystem service benefits in watersheds serving as water sources of large water diversion projects."

Comments:

Little explicit consideration is given to potentially relevant feedback effects of greening the water cycle. This includes the potential for increased local precipitation recycling (P), reduced vapour pressure deficits (VPD), temperatures (T) reduced through increased latent heat flux, which in turn affect water partitioning and thus water yield. These points need at least to be discussed in substantial detail and the conclusion needs to explicitly state these limitations.

Response:

We modeled water yield by considering precipitation and ET, which is a function of NDVI, T, VPD, and radiation. The direct effects of those variables on ET should have been captured by the model, though as suggested by the reviewer, the complex feedbacks and interactions (e.g., increased ET/latent heat flux leading to increased P and reduced surface temperatures) may not have been. Therefore, what we did in this study is solely to disentangle the direct effects of climate and land cover (mainly vegetation greening).

The feedback effects mentioned by the reviewer are indeed not explicitly accounted in our model. We expect some of the feedback effect (e.g., T, P, and VPD) should be captured by the meteorological data we used. Investigating these indirect feedback effects, while very important, is beyond the scope of this study and likely not achievable with the models used here. We realized that the omission of these feedback mechanisms may cause additional uncertainty in modeling results of water supply reduction.

We added the discussion in "4.4 Limitation" in the updated manuscript: "Vegetation greening may in turn affect climate itself, such as through evaporative cooling and moistening of the atmosphere. For example, vegetation greening in the local and upwind area may potentially increase P downwind via increasing atmospheric vapor. This water vapor cycling likely offset some negative vegetation effects of increasing ET on WY. We could not, however, explicitly account for the potential feedbacks between vegetation (NDVI and land cover) and climate (e.g., T, P, and VPD). The climate data used as model inputs would implicitly include a certain level of the feedback effects from vegetation greening, but those effects could not be explicitly disentangled in our analysis. Thus, our results represent an attempt to estimate the direct, first-order net effects of climate and vegetation greening on WY."

Comments:

(2) Linked to (1) is the design of scenarios S2 and S3, which may indeed contain a fallacy. By fixing NDVI and land cover to the values of 2001 (S1) and 2018 (S2), respectively, the authors aim to isolate the "real" and potential effects of greening. This is in principle ok. BUT: it is not clear from the description of the experiment how the related feedbacks in P, T, VPD and even radiation (changes in albedo!) were accounted for. As far as I understood, the assumption for the 2 scenarios was that *only* NDVI and land cover is fixed to the values of the 2 individual years. If this is so, the authors overlook that the observed past P, T, VPD and radiation data already account for changes in NDVI and land cover. What are the effects of that? Do the results then really allow to isolate effects of greening? If I am mistaken here, then I would nevertheless ask the authors to makes this much clearer in the description of their experiment.

Response: Indeed, S2 and S3 are hypothetical scenarios, assuming everything else is the same except the land cover and NDVI. As discussed in the reply above, we did not explicitly account for the feedback effects of T and P as these values are not dynamically coupled in the model, but are driving factors provided externally from existing climate data. The feedback effects on T, P, and VPD should be implicitly included in those input data though they cannot be explicitly decoupled in our model experiments.

To address the reviewer's comment, we conducted new simulation experiments to separate effects of P, T, VPD and radiation. However, all of P, T, VPD and radiation had

insignificant trends during the study period (figure A1). Consequently, their effects on the long-term trends in ET and WY were minor. Therefore, we only showed the combined effects of these climatic variables and did not discuss them separately. In Scenario S2, NDVI and land cover were fixed in 2001, under which changes in WY would be the effects of climate change only. The climate data were derived with in-situ measurements and a climate model, thus they must already contain certain level of feedback effect from a greening world. We cannot remove such effects from the climate data.

Despite these limitations, we still believe that the experiment provided insight on how land cover, greening, and climate each influenced ET and consequently WY. As suggested by the reviewer, we added discussion of the limitations of the modeling in section 4.4:

"……. However, it is impossible to completely decouple the effects of climate change and vegetation greening on ecosystem goods and services due to the tight biophysical interactions and feedbacks between vegetation and climate. The observed vegetation change, as indicated by NDVI, represents the combined effects of changes in climate, human activity (i.e., reforestation, irrigation), and natural vegetation (re)growth. However, because P, T, VPD and radiation had relatively small and insignificant trends during the study period (figure A1), the effects of climate on the vegetation change (NDVI trend and land cover change) were likely relatively minor compared to direct human modification of the landscape and vegetation growth.

Vegetation greening may in turn affect climate itself, such as through evaporative cooling and moistening of the atmosphere. For example, vegetation greening in the local and upwind area may potentially increase P downwind via increasing atmospheric vapor. This water vapor cycling likely offset some negative vegetation effects of increasing ET on WY. We could not, however, explicitly account for the potential feedbacks between vegetation (NDVI and land cover) and climate (e.g., T, P, and VPD). The climate data used as model inputs would implicitly include a certain level of the feedback effects from vegetation greening, but those effects could not be explicitly disentangled in our analysis. Thus, our results represent an attempt to estimate the direct, first-order net effects of climate and vegetation greening on WY."

Comments:

(3) Linked to (2), the description of the models and their actual implementation does not provide sufficient detail to allow the reader to meaningfully assess the results and interpretations. For example, it is completely unclear how NDVI, land cover but also soil data were used. The reader can only assume that these data are somehow used to

estimate the variables PAR and FPAR (Eq.1). Even if this is described elsewhere in detail, it will be necessary to provide this crucial information here as well. In addition, it remains completely opaque, which parameters the two models required and which of those had to be calibrated and which were a priori fixed (e.g. from look-up tables). How were the models calibrated? Was the calibrated model tested on independent data (which should be a rather standard procedure in the year 2021)?

Response:

We added more details and made it clearer in updated version as shown below:

FPAR, R$_s$, T$_s$, and W$_s$ were calculated according to Sims et al. (2005), King et al., (2011), Raich et al., (1991), and Landsberg and Waring, (1997), respectively, as:

$$FPAR = 1.24 \times NDVI - 0.168, \tag{3}$$

$$R_s = 1 - K_1 \times R_a/R_{cs}, \tag{4}$$

$$T_s = \frac{(T-T_{min}) \times (T-T_{max})}{(T-T_{min}) \times (T-T_{max}) - (T-T_{opt})^2}, \tag{5}$$

$$W_s = exp\left(-K_2 \times (VPD - VPD_{min})\right), \tag{6}$$

Where $R_a$ and $R_{cs}$ are respectively actual and clear-sky radiation. The calculation of $R_{cs}$ is based on Raes et al. (2009). $T_{min}$, $T_{max}$, and $T_{opt}$ are respectively the minimum, maximum and optimal air temperatures for photosynthetic activity, varying by biome (derived from the land cover data). $VPD_{min}$ is the minimum VPD exceeding which moisture stress starts to take effect, which also varies by biome, and $K_1$ and $K_2$ are biome-specific empirical parameters that scale the radiation and VPD effects, respectively. The parameters ($\varepsilon_{pot}, T_{min}, T_{max}, T_{opt}, VPD_{min}, K_1$ and $K_2$) are calibrated based on global FLUXNET data through a Monte Carlo simulation (Zhang et al., 2016, 2019).

The Sacramento Soil Moisture Accounting Model (SAC-SMA) was used to model WY in the WaSSI model, driven by ET, precipitation (P), and soil parameters. Soil parameters (see following Table) were generated from multi-layer soil particle-size distribution and depth according to Anderson et al. (2006). The algorithm divides the soil layer into lower and upper zones at different depths and estimates the distribution of moisture—including both tension water components (driven by evapotranspiration and diffusion) and free water components (driven by gravitational forces) in each of these two zones. We add the following Table in Appendix A.

Table: Soil parameter for driving model.

| Parameters types | Parameters | Abbreviations |
|---|---|---|
| Soil Storage Parameters | Upper Layer Tension Water Capacity | UZTWM |
| | Upper Layer Free Water Capacity | UZFWM |
| | Lower Layer Tension Water Capacity | LZTWM |
| | Lower Layer Supplemental Free Water Capacity | LZFSM |
| | Lower Layer Primary Free Water Capacity | LZFPM |
| Baseflow Discharge Rate Parameters | Depletion Rate from LZFPM | LZPK |
| | Depletion Rate from LZFSM | LZSK |
| | Interflow Depletion Rate from UZFWM | UZK |
| Upper to Lower Layer Percolation Parameters | Percolation fraction direct to LZFW | PFREE |
| | Percolation Curve Shape | REXP |
| | Maximum/Minimum Percolation Rate Ratio | ZPERC |

Ref.:

Sims, D. A., Rahman, A. F., Cordova, V. D., Baldocchi, D. D., Flanagan, L. B., Goldstein, A. H., Hollinger, D. Y., Misson, L., Monson, R. K., Schmid, H. P., Wofsy, S. C., and Xu, L.: Midday values of gross CO2 flux and light use efficiency during satellite overpasses can be used to directly estimate eight-day mean flux, 131, 1–12, https://doi.org/10.1016/j.agrformet.2005.04.006, 2005.

King, D. A., Turner, D. P., and Ritts, W. D.: Parameterization of a diagnostic carbon cycle model for continental scale application, 115, 1653–1664, https://doi.org/10.1016/j.rse.2011.02.024, 2011.

Raich, J. W., Rastetter, E. B., Melillo, J. M., Kicklighter, D. W., Steudler, P. A., Peterson, B. J., Grace, A. L., Moore, B., and Vorosmarty, C. J.: Potential Net Primary Productivity in South America: Application of a Global Model, 1, 399–429, https://doi.org/10.2307/1941899, 1991.

Landsberg, J. J. and Waring, R. H.: A generalised model of forest productivity using simplified concepts of radiation-use efficiency, carbon balance and partitioning, 95, 209–228, https://doi.org/10.1016/S0378-1127(97)00026-1, 1997.

Raes, D., Steduto, P., Hsiao, T. C., and Fereres, E.: Aquacrop-The FAO crop model to simulate yield response to water: II. main algorithms and software description, 101, 438–447, https://doi.org/10.2134/agronj2008.0140s, 2009.

Zhang, Y., Song, C., Sun, G., Band, L. E., McNulty, S., Noormets, A., Zhang, Q. and Zhang, Z.: Development of a coupled carbon and water model for estimating global gross primary productivity and evapotranspiration based on eddy flux and remote sensing data, Agricultural and Forest Meteorology, 223, 116–131, https://doi.org/10.1016/j.agrformet.2016.04.003, 2016.

Zhang, Y., Song, C., Band, L. E. and Sun, G.: No Proportional Increase of Terrestrial Gross Carbon Sequestration From the Greening Earth, Journal of Geophysical Research: Biogeosciences, 124(8), 2540–2553, https://doi.org/10.1029/2018jg004917, 2019.

Anderson, R. M., Koren, V. I., and Reed, S. M.: Using SSURGO data to improve Sacramento Model a priori parameter estimates, in: Journal of Hydrology, 103–116, https://doi.org/10.1016/j.jhydrol.2005.07.020, 2006.

Comments:

(4) Linked to (3), no attempt, whatsoever is made, to estimate the uncertainty around the models, their parameters and the associated results. Not even confidence intervals around the regressions (and the underlying parameters) are given. Quite frankly, I find this very surprising, as this should be part of any meaningful and serious scientific protocol.

Response: This is a good point. We validated the model results at multiple scales and provided $R^2$ and RMSE as well as the confidence intervals for the model results based on observed streamflow data (section 3.2 and Fig. 4). As suggested by the reviewer, we now also provide confidence intervals on the regression slopes in our figures (Figs. 3-6) and in the main text.

Comments:

Additional specific comments:

2, l.42: this is a completely non-sensical use of the term "drought". The term "drought" is always refers to a negative anomaly with respect to a specific local reference value, defining a "normal", typically a median. By convention, conditions below this normal are then defined as "drought". By extension, there can then be no location with more "frequent" droughts than other locations, as drought is always the deviation from the local/regional normal.

Response: We rephrased the sentence as "the UHRB is quite vulnerable to hydrological drought events". There are many drought indices to identify drought with different criteria (Hayes et al., 2002). In general, hydrological drought refers to a severe lack of water in the hydrological system, manifesting in abnormally low streamflow in rivers and abnormally low water levels in lakes, reservoirs, and groundwater (van Loon, 2015). Although there are, by definition, half of periods with water yield below the average (or median) for all areas, not all incidents with streamflow below the average can be called drought events. Some areas have unstable water yield with greater fluctuations than others. Therefore, these areas generally experience longer and more severe drought events.

Ref.,

Hayes, M. J., Alvord, C. and Lowrey, J.: Drought indices, National Drought Mitigation Center, University of Nebraska., 2002.
van Loon, A. F.: Hydrological drought explained, Wiley Interdisciplinary Reviews: Water, 2(4), 359–392, https://doi.org/10.1002/wat2.1085, 2015.

2, l.50-51: if afforestation was meant to safeguard water availability, then this is in contradiction with l.55-56. Please rephrase.

Response: The statement was rephrased as "safeguard water quality and increase soil water storage".

2, l.56: greater leaf area in itself does of course not increase transpiration. Vegetation metabolic activity increases transpiration. Leaf area is merely an indicator for increased metabolic activity and thus transpiration.

Response: Revised "greater leaf area" as "vegetation greening".

2, l.61: should read as "…are not feasible…"

Response: Changed as suggested. Thanks!

3, l.69: what are "hydrological entities"?

Response: Revised as "hydrological variables".

3, l.76: droughts are low frequency phenomena that require considerable time to develop and to recede. The 18 years of this study are thus likely not enough to make a meaningful statement about changes in drought regimes.

Response: Revised as "hydrological drought risks".

4, Figure 1: please also show the location of the reservoirs and hydroelectric facilities.

Response: We added a legend to mark the location of the hydroelectric facilities as is shown in following fig. (Fig 1).

[Figure]

6, l.130-131: irrelevant, can be omitted

Response: Removed.

6, l.132-133: not clear what is meant here. Which other models?

Response: We now refer to more specific models in the manuscript: the Penman-Monteith (Penman, 1948; Monteith, 1965), RHESSys (Tague and Band, 2004), ORCHIDEE (Krinner et al., 2005) models etc.

Ref.:

Krinner, G., Viovy, N., de Noblet-Ducoudré, N., Ogée, J., Polcher, J., Friedlingstein, P., Ciais, P., Sitch, S., and Prentice, I. C.: A dynamic global vegetation model for studies of the coupled atmosphere-biosphere system, https://doi.org/10.1029/2003GB002199, 1 March 2005.

Monteith, J. L.: Evaporation and environment, 1965.

Penman, H. L.: Natural evaporation from open water, bare soil and grass, 193, 120–145, https://doi.org/10.1098/rspa.1948.0037, 1948.

Tague, C. L. and Band, L. E.: RHESSys: Regional Hydro-Ecologic Simulation System—An Object-Oriented Approach to Spatially Distributed Modeling of Carbon, Water, and Nutrient Cycling, 8, https://doi.org/10.1175/1087-3562(2004)8<1:RRHSSO>2.0.CO;2, 2004.

6, l.138: more detail is needed for this choice here. Why 45%? How sensitive is the model to this choice?

Response: The percent of PAR in the total solar radiation indeed varies slightly from place to place. We took the value based on the published value from Running et al., (2000). However, because this parameter is a simple scalar, a change in this parameter would not influence the interannual variation of evapotranspiration or streamflow since the other parameters in the LUE model were optimized to maximize fit between the measured and modeled ET (i.e., any change in the percentage of shortwave radiation that is PAR would be proportionally offset by adjustments in the calibrated light-use efficiency).

Running, S. W., Thornton, P. E., Nemani, R., and Glassy, J. M.: Global Terrestrial Gross and Net Primary Productivity from the Earth Observing System, in: Methods in Ecosystem Science, Springer New York, 44–57, https://doi.org/10.1007/978-1-4612-1224-9_4, 2000.

6, l.139-140: how were PAR and FPAR determined?

Response: PAR was calculated as 45% of the total shortwave radiation; FPAR was calculated based on NDVI (Sims et al. 2005):

$$FPAR = 1.24 \times NDVI - 0.168$$

We have further clarified this in the manuscript.

Ref.

Sims, D. A., Rahman, A. F., Cordova, V. D., Baldocchi, D. D., Flanagan, L. B., Goldstein, A. H., Hollinger, D. Y., Misson, L., Monson, R. K., Schmid, H. P., Wofsy, S. C., and Xu, L.: Midday values of gross CO2 flux and light use efficiency during satellite

overpasses can be used to directly estimate eight-day mean flux, 131, 1–12,
https://doi.org/10.1016/j.agrformet.2005.04.006, 2005.

6, l.139-150: Much more detailed is needed on which parameters these models feature
and how the parameters were determined, including their prior distributions and the
calibration strategy applied.

Response: We have clarified which parameters were tunable and how they were
calibrated (see previous responses above).

7, l.163: R2 and NSE have a very similar information content: NSE collapses to R2 in
the absence of a bias. Thus, I am not sure of the added value of using R2 as
performance metric here.

Response: This is a good point, though the initial intention of using both $R^2$ and NSE was
to provide more information to readers. Since bias may indeed be present, we have
decided to show both of them, and many readers may not be as familiar with NSE as $R^2$.

7, l.164: reliable? Many would argue otherwise (e.g. Schaefli and Gupta, 2007, HP). In
addition, what does "reliable" actually mean here?

Response: We thank the reviewer for pointing this out. The word "reliable" was removed.

10, Figure 4: given that the model only provides monthly estimates of water yield, the
model does not do a particularly good job in reproducing the observed water yield, in
particular for the 2012-2014 period. What is the implication of this? What are the
uncertainties around that? How does it affect the results and interpretation?

Response: The inflow of the Danjiangkou Reservoir is controlled by dams on the
mainstream Han River. Since the flow records of hydropower plants in China is not
publicly accessible, to what extent the inflow of the Danjiangkou Reservoir can represent
the WY of overall UHRB is unknown. However, we think the effects of bias of WY
during 2012-2014 were limited. The measured and simulated WY not only have good
correlation but also have nearly identical trends (see the following figure). This situation
has been stressed in "2.4 Model Evaluation", and we added statement about the
comparison of trend of measured and simulated WY in the model evaluation section.

[Figure]

11, l.225-226: I am concerned that the change point analysis here is really very sensitive to the rather short time period considered and that the points identified here may be mere artefacts (e.g., Zhou et al., 2019; HSJ). I strongly suggest to omit this from the analysis.

Response: As suggested, we have removed this from the analysis.

11, l.220ff: am I right to assume that scenarios S1 and S2 are shown and discussed in this section? Please clarify and make this explicit.

Response: Yes, we have clarified this in the revised manuscript.

12, 237ff: not clear what is considered here. Is it the difference between S1 and S2? If yes, I wonder how much of the correlation is spurious, as NDVI is kept constant, while still using observed T and VPD that are the result of a variable vegetation cover. This needs to be made much clearer.

Response: Yes, it is the difference between S1 and S2. We have made this clearer in the revised manuscript.

---

## Author Comment (AC3)

**To Reviewer #2,**

Zhang et al (2021) coupled the CCW and WaSSI models to study how vegetation greening impacted water yield of the Upper Han River Basin (UHRB). They first simulate water yield change from 2001-2018 to evaluate the model. Afterwards, they run two simulations to isolate the effect of vegetation on water yield and the effect on future potential water yield. Zhang et al (2021) show that vegetation greening significantly reduced water yield. The water yield reduction was stronger during warm or dry years. Furthermore, they show that greening could increase the number of droughts. They discuss their results in relation to the important role of the UHRB to provide water to other regions through a diversion project.

The study has an easy to understand set-up and addresses a relevant subject. The manuscript is clearly written. I listed some (major and minors) comments and suggestions below, both on the content and text.

Response: We appreciate the positive comments and summary of this study. Please find our specific responses to each of the comments below (in blue).

Comments:

The authors show that vegetation greening significantly reduced water yield and streamflow during the last decades. The authors discuss the implications for the SNWDP and other Water Diversion Projects and state that (future) vegetation greening could potentially reduce the annual water yield supply by 7.3 km$^3$. A few processes are missing in the manuscript that impact streamflow under changing vegetation. These processes could reduce the 'negative' effects of vegetation greening on water yield. First, the extra evaporated water will partly recycle back to the Earth's surface and increase precipitation (P) (potentially within the UHRB catchment). This could have impacted your P during the studied years (therefore, the S2 and S3 scenarios are not entirely independent of vegetation status), and likely has an impact on future water yield. The study cannot separate this effect on increased P, but they could at least be included in the discussion of the manuscript.

Response: We appreciate the insightful comments here. First, we agree that effect of precipitation recycling feedback on WY was not captured in our modeling. The P used in this study (derived from a combination of model and ground data) was not generated by our model, but it was used as a driver to the model. If there is a such climatic feedback, it would already be implicitly included in the observed P data, but our model would not be able to disentangle this feedback.

We found that P did not have a significant increasing trend during the period, though ET increased significantly. We recognize that increasing ET may induce increasing P locally or downwind. However, the P in the UHRB is more greatly influenced by the Asia Monsoon, which comes from the Pacific Ocean from south-eastern China. The significant and widespread greening was also observed in south-eastern China (Chen et al., 2019) and transferred a larger amount of moisture which might be brought to the UHRB and

potentially induce a P increase. However, these effects are difficult to quantify. Even if we know how much P in the UHRB was from the upwind area, it is beyond the capacity of our model to quantify the amount of P that comes from the extra ET from greening. We believe that the effects of greening in local and upwind area on P in the UHRB were limited. Roughly 67% of P in the Yangtze River Basin (where the UHRB located) comes from the Pacific Ocean (Tuinenburg et al., 2020), and a recent modelling study found that vegetation greening only induced a P increase of 1.5% per decade in the Yangtze River Basin from 1982 to 2011 (Li et al., 2018).

However, we agree with the reviewer that this is an important limitation, and we have therefore expanded discussion of this potentially important feedback in the Discussion section"4.4 Limitation".

Ref.:

Chen, C., Park, T., Wang, X., Piao, S., Xu, B., Chaturvedi, R. K., Fuchs, R., Brovkin, V., Ciais, P., Fensholt, R., Tømmervik, H., Bala, G., Zhu, Z., Nemani, R. R. and Myneni, R. B.: China and India lead in greening of the world through land-use management, Nature Sustainability, 2(2), 122–129, https://doi.org/10.1038/s41893-019-0220-7, 2019.

Tuinenburg, O. A., Theeuwen, J. J. E., and Staal, A.: High-resolution global atmospheric moisture connections from evaporation to precipitation, 12, 3177–3188, https://doi.org/10.5194/ESSD-12-3177-2020, 2020.

Li, Y., Piao, S., Li, L. Z. X., Chen, A., Wang, X., Ciais, P., Huang, L., Lian, X., Peng, S., Zeng, Z., Wang, K. and Zhou, L.: Divergent hydrological response to large-scale afforestation and vegetation greening in China, Science Advances, 4(5), eaar4182, https://doi.org/10.1126/sciadv.aar4182, 2018.

Second, the rising $CO_2$ concentrations are expected to increase the water use efficiency of vegetation, and this could reduce the 'negative' effects of future afforestation.

 Response:

We thank the reviewer for the comments on the effects of $CO_2$ on water use, ET, and the water balance. There is no clear consensus in the literature on the extent to which increasing $CO_2$ concentrations will affect ET at the ecosystem level even with an increase in leaf level water use efficiency (see e.g., Frank et al., 2015; Ward et al., 2018), but it clearly has the potential to affect future forest productivity and water use. One of our model limitations is that we cannot directly evaluate the $CO_2$ effects on water use efficiency and thus ET and streamflow. However, we may have partially captured the effect because the underlying water use efficiency used in the model was calibrated based on global flux tower data, and over the short study period evaluated in this study (2001-

2018), $CO_2$ concentrations likely did not change enough to have a strong effect; Frank et al. (2015), for example, found that $CO_2$ fertilization led to only ~20% increases in evergreen forest WUE over the entire 20th century.

In response to this comment, we have provided further discussion about the potential $CO_2$ fertilization effect in section 4.4 ("Limitations"): "Future change in both temperature and precipitation will likely reduce the water supply in the UHRB, and increasing atmospheric $CO_2$ is also likely to enhance vegetation greening due to the $CO_2$ fertilization effects. Such greening effects could increase ecosystem productivity and water use efficiency, and thus alter the water cycle. However, the effects $CO_2$ fertilization on total water use (ET) may not decrease as much as previously thought (Ward et al., 2018) and can be uncertain because $CO_2$ fertilization may also cause an increase in total forest leaf area and a shift in plant species, both of which would also affect ET. While we the potential effects of $CO_2$ fertilization on WY were not explicitly included in this study due to the structure of the light-use efficiency model, previous work has shown that forest water-use efficiency only increased by about 15-20% over the entire 20th century (Frank et al., 2015) and such effect weakened recently (Wang et al. 2021), so the effects of CO2 fertilization on WY over our comparatively short 2001-2018 study period were likely quite small.".

Frank, D. C. et al. (2015), Water-use efficiency and transpiration across European forests during the Anthropocene, Nature Climate Change, 5, 579-583.

Ward, E.J., Oren, R., Seok Kim, H., Kim, D., Tor-ngern, P., Ewers, B.E., McCarthy, H.R., Oishi, A.C., Pataki, D.E., Palmroth, S. and Phillips, N.G., 2018. Evapotranspiration and water yield of a pine-broadleaf forest are not altered by long-term atmospheric [CO2] enrichment under native or enhanced soil fertility. Global change biology, 24(10), pp.4841-4856.

Wang et al. 2021. Recent global decline of CO2 fertilization effects on vegetation photosynthesis, Science, 370: 1295-1300.

L31: For example … hydrological services: this sentence should be rewritten

Response: It was revised as "However, the sustainability of such projects depends on water supply from the donor watersheds, which is uncertain due to rapid vegetation greening and climate change."

L57: Consume instead of consumes

Response: Revised.

L61: are instead of is

Response: Revised.

Please adjust figure 1a (inset) to meet the HESS guidelines (remove the dashed line south of China to depoliticise the manuscript)

Response: Revised.

L136: please specify that is the light use efficiency.

Response: Revised.

L140: how are the values of the 'environmental scalars' determined? Are they independent of the vegetation data? And how is APAR determined? Is APAR also fixed under de S2 and S3 scenario?

Response: We made it clearer in revised manuscript as shown belong:

APAR is the product of FPAR and PAR. PAR is taken as 45% of shortwave radiation (Running et al. 2000). FPAR, $R_s$, $T_s$, and $W_s$ were calculated according to Sims et al. (2005), King et al., (2011), Raich et al., (1991), and Landsberg and Waring, (1997), respectively, as:

$$FPAR = 1.24 \times NDVI - 0.168, \tag{3}$$

$$R_s = 1 - K_1 \times R_a / R_{cs}, \tag{4}$$

$$T_s = \frac{(T - T_{min}) \times (T - T_{max})}{(T - T_{min}) \times (T - T_{max}) - (T - T_{opt})^2}, \tag{5}$$

$$W_s = exp \left(-K_2 \times (VPD - VPD_{min})\right), \tag{6}$$

Where $R_a$ and $R_{cs}$ are respectively actual and clear-sky radiation. The calculation of $R_{cs}$ is based on Raes et al. (2009). $T_{min}$, $T_{max}$, and $T_{opt}$ are respectively the minimum, maximum and optimal air temperatures for photosynthetic activity, varying by biome (derived from the land cover data). $VPD_{min}$ is the minimum VPD exceeding which moisture stress starts to take effect, which also varies by biome, and $K_1$ and $K_2$ are biome-specific empirical parameters that scale the radiation and VPD effects, respectively. The parameters ($\varepsilon_{pot}, T_{min}, T_{max}, T_{opt}, VPD_{min}, K_1$ and $K_2$) are calibrated based on global FLUXNET data through a Monte Carlo simulation (Zhang et al., 2016, 2019).

Ref.:

Sims, D. A., Rahman, A. F., Cordova, V. D., Baldocchi, D. D., Flanagan, L. B., Goldstein, A. H., Hollinger, D. Y., Misson, L., Monson, R. K., Schmid, H. P., Wofsy, S. C., and Xu, L.: Midday values of gross CO2 flux and light use efficiency during satellite overpasses can be used to directly estimate eight-day mean flux, 131, 1–12, https://doi.org/10.1016/j.agrformet.2005.04.006, 2005.

King, D. A., Turner, D. P., and Ritts, W. D.: Parameterization of a diagnostic carbon cycle model for continental scale application, 115, 1653–1664, https://doi.org/10.1016/j.rse.2011.02.024, 2011.

Raich, J. W., Rastetter, E. B., Melillo, J. M., Kicklighter, D. W., Steudler, P. A., Peterson, B. J., Grace, A. L., Moore, B., and Vorosmarty, C. J.: Potential Net Primary Productivity in South America: Application of a Global Model, 1, 399–429, https://doi.org/10.2307/1941899, 1991.

Landsberg, J. J. and Waring, R. H.: A generalised model of forest productivity using simplified concepts of radiation-use efficiency, carbon balance and partitioning, 95, 209–228, https://doi.org/10.1016/S0378-1127(97)00026-1, 1997.

Raes, D., Steduto, P., Hsiao, T. C., and Fereres, E.: Aquacrop-The FAO crop model to simulate yield response to water: II. main algorithms and software description, 101, 438–447, https://doi.org/10.2134/agronj2008.0140s, 2009.

Zhang, Y., Song, C., Sun, G., Band, L. E., McNulty, S., Noormets, A., Zhang, Q. and Zhang, Z.: Development of a coupled carbon and water model for estimating global gross primary productivity and evapotranspiration based on eddy flux and remote sensing data, Agricultural and Forest Meteorology, 223, 116–131, https://doi.org/10.1016/j.agrformet.2016.04.003, 2016.

Zhang, Y., Song, C., Band, L. E. and Sun, G.: No Proportional Increase of Terrestrial Gross Carbon Sequestration From the Greening Earth, Journal of Geophysical Research: Biogeosciences, 124(8), 2540–2553, https://doi.org/10.1029/2018jg004917, 2019.

L161: the streamflow records of the reservoir ( / the Danjiangkou Reservoir)

Response: Revised.

L173: dynamic greening effects instead of dynamics greening effects

Response: Revised.

L179: The Mann-Kendall test is used for trend and change point detection. Could the authors elaborate on the change points you found? Why did they decided to use change-point detection analyses instead of trend analyses only? What extra information do these change-points add to the discussion or results of the manuscript?

Response: We used the change-point detection because we found the WY did not have a statistically significant trend, then want to get another method to detect the WY change. As reviewer #1 suggested, the change point detection related content was removed since the 18-year study period is quite short for the change point method, and its results are sensitive to outliers.

Fig 4a+b legend: km$^3$ per year / mm per year.

Response: Revised.

Fig 4a: the simulated WY seems to show a higher decreasing trend than the measured WY. Was there a negative trend in the measured WY?

Response: They have almost identical trends (see the following fig). The trend lines are added in the Figure 4a.

[Figure]

She et al, 2017 (fig 2a) (https://doi.org/10.1002/2016JD025702) fitted an increasing trend through WY at the Danjiangkou Reservoir between 2000 and 2010 (same data). How does this compare to your results?

Response: Indeed, the measured data also showed an increasing trend in WY during 2000-2011 with a slope of 1.45 km$^3$ per year (see the following Figure), which is consistent with She et al's (2017) data. However, this appears to be largely an effect of having abnormally low values early in the record and anomalously high WY values in

2009 and 2010. The longer time series in our study shows a decrease in WY since 2010 leading to overall negative trends in WY from 2001-2018.

[Figure]

Figure: Temporal variation of observed inflow of the Danjiangkou Reservoir. The blue and red dashed lines are the trend line of 2001-2011 and 2001-2018.

L237: Why did WY increase due to vegetation greening in high elevation areas? Could it also be a climate-related effect in these high-elevation regions?

Response:

Here, "high elevation" refers to 3000 m and above, which only occupied a small proportion of the study area. There may be several reasons for the increase in WY here. First, the greening trend in high elevation was not as strong as in the low elevation (see the Figure below). Therefore, the decrease in WY in the high elevation was small (less than 15 mm in 18 years). Second, the annual mean temperature at the high elevation area is only around 3 °C (see the following Figure). The low temperature may greatly limit vegetation activity, even if vegetation is greening. Therefore, greening had limited effects on ET in the high elevation, but can increase soil water capacity, thus increasing WY.

[Figure]

Line 239-240: Did you mean to refer to fig. 5c instead of 5b?

Response: It is Figure 5c. Revised.

8: How is the relative change calculated? Relative to the year 2010, or the S2 scenario? Why is the sign of the absolute WY change opposite of that of the relative WY change. What does this say about the effects of greening versus climate?

Response: The relative changes refer to the proportion of WY (ET) changes to those in the scenario without greening (S2). We have made this clearer in the revised manuscript.

In the dry period, the greening effects on WY were lower in magnitude than those of the wet period because of the soil moisture limitation. Consequently, the WY change from greening in magnitude had positive correlation with P. However, the WY has resilience to short term drought thanks to soil water storage. Therefore, the WY change in proportion was less than that of P change. Moreover, as conditions get wetter, the negative effects of greening on water yield will increase before reaching the limitation of vegetation activity and energy supply. Thus, more P will not induce a comparable magnitude of WY increase. As a result, the magnitude of WY relative change from greening will decrease with increasing P within the range of climate variation we encountered.

L261: 2001-2018 instead of 2001~2018

Response: Revised.

L307: 'Unlike the Loess Plateau … but climate did' seems to contradict with your results. How should this sentence be interpreted?

Response: We meant to state that climate masked the effects of greening. From the following Figure (Figure 5c in the manuscript), vegetation greening in the UHRB significantly reduced WY, but WY did not decrease significantly as a result of climate variability. To make it explicit, this sentence was revised as "Unlike the previous two examples, vegetation greening in the UHRB induced a substantial decrease in WY (scenario S2), but WY under the combined effects (scenario S1) did not have a statistically significant trend as a result of the large interannual variation of climate in UHRB." The sentence was also moved to the end of the paragraph as a lead to the following paragraph.

---

## Author Comment (AC4)

To Reviewer #3,

The manuscript simulated and analysed the effect of vegetation greening on water yield for the South-to-North diversion project (SNWDP). The manuscript is presented in a clear way and method and analyses are internally logical and consistent. However, the manuscript's narrow scope and sole focus on 'watershed management' does not account for critically important connectivities in the water cycle over land, or discuss the overall sustainability benefits or trade-offs of land and water management options. While limitations in scope are necessary to all scientific studies, here, the limitations and the insufficient acknowledgements thereof have resulted in potentially misleading statements about the effect of vegetation greening on runoff and lead to misinterpretations in terms policy/management implications.

Response: We thank the reviewer for the insightful comments, and we totally agree that we should provide a balanced account of the ecosystem goods and services provided by forests rather than solely focusing on watershed management. We recognize the multiple positive ecosystem goods and services associated with vegetation greening, but we also think it is important for policymakers to understand the feedbacks and potential unintended consequences from greening on water yield. We also agree that we should make the limitations of this study clearer and more explicit. We have addressed the reviewer's concerns in the revision, as discussed in detail below.

The authors write for example: "Overall, our study suggests that afforestation could potentially reduce local WY, thus weakening the capacity of the water supply to SNWDP." and "Our study suggests that improved watershed management (e.g., forest management and reducing water use) is needed to address the effects of vegetation greening and climate change on water supply capacity in watersheds serving as water sources for large water diversion projects."

The authors might not mean this, but it is easy to interpret this as an argument for limiting re-greening and reducing vegetation. The slight absurdity in such sentence formulations can perhaps be illustrated by applying the same logic to the model simulation results of (Kleidon, Fraedrich, and Heimann 2000), who found that a global 'desert world' yields 37 000 $km^3$ per year runoff whereas a 'maximal green world' yields 28 000 $km^3$ per year of runoff. Of course, Kleidon et al., (2020) also noted that both precipitation over land and total evaporation from land were substantially higher in the 'green world' scenario. However, with the authors' logic and narrow focus on 'water yield', would they have stated that 'the presence of terrestrial vegetation potentially weakens the capacity of the water supply'?

I recommend the authors to (1) either expand their scope (to test how the results would be affected by accounting for moisture recycling including greening in upwind areas, and/or $CO_2$ fertilization under different assumptions), or (2) substantially revise the framing and conclusions of the paper. To test the sensitivity of the results to moisture recycling and greening in upwind moisture supply areas, the authors could for example make use of publicly available data of atmospheric moisture flows (Tuinenburg and Staal 2020; Tuinenburg, Theeuwen, and Staal 2020; Link et al. 2020). Sensitivities to $CO_2$ fertilization could potentially be investigated by testing different parameterizations in the models.

If option 1 is considered out of the scope, I recommend the authors to revise the title, the abstract, discussion, and conclusions so it is among others clear that i. the vegetation

change considered are only within the basin; ii. that key processes and feedbacks such as moisture recycling are missing from the simulations which are likely to reduce the water yields risks reported (see for example Weng et al. (2019) that shows that strategic location of reforestation in upwind areas can in fact help support water use demands, and Wang-Erlandsson et al. (2018) that shows that irrigation in India and other countries contributes to precipitation over China by increased moisture supply); and iii. that overall ecosystem services and trade-offs (e.g., Onaindia et al. 2013) provided by reforestation or restoration projects have not been considered herein (please consider discussing these). I find the authors' current recommendations to be cautious of over-reliance on the water supply of the SNWDP project under future greening scenarios to be motivated and relevant. The authors could also elaborate on what they mean by consideration of 'forest management', for example referring to examples of reforestation approaches that provide relatively high ecosystem service benefits with low evaporation rates. Elaborating on these points could help make the paper more nuanced and insightful, and less prone to being mis-interpreted.

**References**

Kleidon, Axel, Klaus Fraedrich, and Martin Heimann. 2000. "A Green Planet Versus a Desert World: Estimating the Maximum Effect of Vegetation on the Land Surface Climate." Climatic Change 44 (4): 471–93.

Link, Andreas, Ruud van der Ent, Markus Berger, Stephanie Eisner, and Matthias Finkbeiner. 2020. "The Fate of Land Evaporation – a Global Dataset." Earth System Science Data. https://doi.org/10.5194/essd-12-1897-2020.

Onaindia, Miren, Beatriz Fernández de Manuel, Iosu Madariaga, and Gloria Rodríguez-Loinaz. 2013. "Co-Benefits and Trade-Offs between Biodiversity, Carbon Storage and Water Flow Regulation." Forest Ecology and Management 289 (February): 1–9.

Tuinenburg, Obbe A., and Arie Staal. 2020. "Tracking the Global Flows of Atmospheric Moisture and Associated Uncertainties." Hydrology and Earth System Sciences 24 (5): 2419–35.

Tuinenburg, Obbe A., Jolanda J. E. Theeuwen, and Arie Staal. 2020. "High-Resolution Global Atmospheric Moisture Connections from Evaporation to Precipitation." Earth System Science Data 12 (4): 3177–88.

Wang-Erlandsson, L., Ingo Fetzer, Patrick W. Keys, Ruud J. van der Ent, Hubert H. G. Savenije, and Line J. Gordon. 2018. "Remote Land Use Impacts on River Flows through Atmospheric Teleconnections." Hydrology and Earth System Sciences 22 (August): 4311–28.

Weng, Wei, Luís Costa, Matthias K. B. Lüdeke, and Delphine C. Zemp. 2019. "Aerial River Management by Smart Cross-Border Reforestation." Land Use Policy.

We thank the reviewer for the clear, insightful, and constructive comments and suggestions.

First, we need to explicitly state that we do not advocate limiting greening and reducing forests. We do not recommend that afforestation should be halted or stopped in this region, because afforestation has many other benefits despite reducing water yield. As discussed in the manuscript, afforestation significantly reduced sediment and improved water quality in this region. However, what we stressed is that there is a tradeoff between

water quality improvement and water resource availability for water diversion projects. Our study was indeed conducted from the perspective of water yield alone. What we found is a risk of water supply reduction due to afforestation and thus a need for comprehensive watershed management to deal with such a risk and to manage that tradeoff.

We realize that we did not deliver as clear a message as we intended about the effects of afforestation in the region thanks to the reviewers' comments. As a result, we have revised the manuscript according to the reviewer's second suggestion.

1) Introduction. We substantially reframed the second and third paragraph of the Introduction. We talk about the afforestation (greening) induced tradeoff ecosystem services. We did not narrowly stress the negative effects of afforestation (greening) here, but first stated the broader benefits of afforestation on reducing sediment in the streamflow and improve water quality, as well as other ecological benefits. Then we talk about the water supply capacity to the water diversion project, and indicated the amplified uncertainties in water availability due to afforestation (greening).

2) We made the following revisions in the Discussion:

   a) In "4.1 Reduced water yield and drought amplification from greening", we discussed the greening effects more conservatively. Specifically, we stressed that "However, as the latest round afforestation efforts are about to conclude, implementation of afforestation projects will likely soon slowdown in China, and combined with potential limitations from future water stress, the greening trend will not likely continue at their same levels in the UHRB." And the discussion about the future leaf area increases was removed.

   b) The "4.2 Reduced capacity of water supply to SNWDP from greening" was revised as "Trade-offs among ecosystem goods and services induced by greening" and its content was reframed. Specifically, to emphasize the "trade-off" instead of water availability alone, we substantially discussed water quality improvement, as well as other ecosystem goods and services due to greening in the first paragraph of 4.2. Then we shortened the discussion on how greening reduced water availability and moved the discussion about water supply capacity to the water diversion project to section 4.3.

   c) In "4.3 Implications for water diversion projects", we provided more discussion on watershed and forest management. Specifically, we moved the discussion about water supply capacity to the water diversion project from 4.2 to combine with the initial first paragraph of 4.3 to discuss the water supply concerns of

water diversion projects. Then we offered more specific recommendations for alleviating effects of greening on water supply. For example, we suggest that using natural regeneration with local tree species rather than artificial plantations to control erosion and conduct ecological restoration. Forest management such as stand thinning to reduce water use and fire risk and increase resilience of the ecosystem should be also considered as part of the integrated watershed management.

d) We added "4.4 Limitations" to discuss the limitations of this study in terms of experiments and external impact factors. We first discussed the possible influences of the interaction between vegetation and climate due to limitations of the experimental design. We then substantially discussed the uncertainty from possible moisture recycling. The enhancement of local ET had the potential to increase P. However, the P in the UHRB is primarily controlled by the Monsoon, which comes from the Pacific Ocean in southeast. The significant and widespread greening was also observed in the southeast China and evaporated a larger amount of moisture which might be brought to the UHRB and potentially induce a P increase. If this happens, such precipitation should already been capture by the precipitation data. In the last part of 4.4, we discussed the possible effects of increasing atmospheric $CO_2$ on water cycle.

3) Accordingly, we revised the conclusion to stress the "trade-off" and suggest that the "improved watershed management (e.g., forest management and reducing water use) is needed to maximize the ecosystem service benefits in watersheds serving as water sources of large water diversion projects."

As to the option 1 suggested by the reviewer, this would be a great direction for future studies as we believe that it will be interesting to investigate the effects of greening in the upwind area on regional precipitation and the hydrologic cycle. However, as recognized by the reviewer, it is beyond the scope and capacity of this study, mainly because of the following challenges: 1) As Tuinenburg et al., (2020) found, the only 33% of P in the Yangtze River Basin (where the UHRB located) is evaporated from land. Even if we know how much P in the UHRB was from the upwind area, it is challenging to find the amount of P that comes from the extra ET from greening. 2) There is a lack of high-resolution moisture data currently. Tuinenburg et al., (2020) developed a high-resolution (0.5°×0.5°) global atmospheric moisture product, but its resolution is still too coarse for watershed studies (like UHRB). It would bring significant uncertainties to use the 0.5° resolution data in a 250 m-resolution study.

Ref.:

Cao, S., Sun, G., Zhang, Z., Chen, L., Feng, Q., Fu, B., …& Wei, X. (2011). Greening China Naturally. Ambio, 40(7), 828-831.

Tuinenburg, Obbe A., Jolanda J. E. Theeuwen, and Arie Staal. 2020. "High-Resolution Global Atmospheric Moisture Connections from Evaporation to Precipitation." Earth System Science Data 12 (4): 3177–88.

---

## Author Response (AR2)

**Dear Editor,**

**We are submitting a revised version of the manuscript "Vegetation Greening Weakened the Capacity of Water Supply to China's South-North Water Diversion Project" for your consideration in publishing in HESS.**

**We have addressed all concerns of the reviewer and associate editor. Key revisions include:**

1) **We substantially stressed the benefits of forest service associated with vegetation greening against the potential strong water consumption revealed by our study in Abstract, Discussion and Conclusion. The Discussion section 4.1 were expanded to discuss the atmospheric moisture transport and the precipitation recycle effects on our results.**

2) **To avoid confusion and better reflect our finding, we adjusted our previous title "***Vegetation Greening **Significantly Reduced** the Capacity of Water Supply to China's South-North Water Diversion Project***" to "***Vegetation Greening **Weakened** the Capacity of Water Supply to China's South-North Water Diversion Project***"**

3) **We addressed all minor comments of the reviewer and carefully re-edited this manuscript on presentation and grammar.**

**Please see our point-to-point responses to the reviewers below. We hope that the revision meets the high standard of HESS.**

**Sincerely,**

**Corresponding authors**

**To Reviewer #1,**

Comments:

The authors have invested quite some effort to address the reviewer comments. As a result, the revised manuscript has improved, in particular with respect to the detail provided in the description of the experiment and the discussion of the limitations. I commend the authors for that.

Response: We appreciate your recognition of our previous revision very much. Your insightful comments and suggestions helped us a lot on improving the manuscript. Please find our point-by-point response below.

However, this being such an important topic, it is crucial to realize that the manuscript has embarked on the challenging necessity to walk on a very thin line between the immediate and direct services of water yield and the larger-perspective services related to a changing climate. As such, it is important that the manuscript provides a balanced view of the potential trade-offs involved. Although the authors promised to do so in their author responses, I still feel that the message still heavily leans towards the importance of water

yield and that it is not yet sufficiently balanced towards other services. In particular, in the abstract and the introduction, very little to no specific mention is made of the potential of afforestation for example to increase CO2 sequestration and to thus provide carbon sinks. Similarly, the importance and relevance of forests as atmospheric moisture source that sustains downwind precipitation as well as their role for (evaporation) cooling have not yet been given sufficient weight neither in the introduction nor in the discussion (e.g. section 4.2) and conclusion sections of the manuscript. While I appreciate the authors efforts to explicitly mention trade-offs, more specific detail is needed to avoid a very one-dimensional message to the reader, which may have very wide-reaching consequences. I thus strongly encourage the author to invest some more effort to provide a really balanced message.

Response: Done as suggested. We fully realize the potential misunderstanding of ecosystem services provided by reforestation against the water consumption revealed by our study. In this revised version, we added emphasis on the ecosystem goods and services that vegetation greening can bring both in the Abstract the Conclusions sections, and further discussed the potential feedbacks among vegetation, evapotranspiration, and precipitation in the Discussion.

Specifically, to stress the benefit of forest service in the Abstract (lines 26-28), we added "Although vegetation greening can bring enormous ecosystem goods and services (e.g., carbon sequestration & water quality improvement), it could aggravate the severity of hydrological drought. Our analysis indicated that vegetation greening in UHRB reduced about a quarter of water yield on average during drought periods.".

To discuss the potential feedback of vegetation to precipitation, we added a new paragraph in Section 4.1 (lines 345-361): "Although vegetation greening had negative effects on WY at the regional scale as we found in the UHRB, some previous studies indicated that vegetation greening could increase WY at larger spatial scales. Enhanced ET from vegetation could moisten the atmosphere, thus drive P to increase. On the other hand, Makarieva et al., (2007) argued that increases in ET could drive the transport of water vapor across continental space via changing atmospheric pressure dynamics. Therefore, vegetation greening in the UHRB may also potentially increase P via increasing atmospheric vapor. However, the amount of recycled P is strongly dependent on the watershed area, with larger geographical expanses having greater potential for recycling. At the regional scale, 87% of the atmosphere moisture through enhanced ET in nine large basins around the world are not likely to recycle back as P, but transport to other regions. The proportion of recycled P in the UHRB would be lower due to its smaller extent. Tuinenburg et al., (2020) found that moisture from ocean contributes approximately 67% of P in the Yangtze River Basin (where the UHRB located). In addition, the atmosphere moisture through enhanced ET in the upwind areas was possible to transport to the downwind, thus increase P. The upwind areas of the UHRB, Southeast China, also experienced vegetation greening simultaneously, which may lead to P

increase in the UHRB. A modelling study found that vegetation greening only induced an P increase of 1.5% per decade in the Yangtze River Basin from 1982 to 2011. Given that no particular trend in annual P in the UHRB was observed (Figure A1b), the effects of local and upwind greening on P may be limited during the study period. However, such complex feedbacks among vegetation, evaporation and precipitation are worth investigating in the future.".

To strengthen the trade-off among ecosystem goods and services induced by greening, we expanded expressions in Section 4.2.

In the conclusion, we further reflect forest service by adding "Despite the enormous ecosystem goods and services provided by forest (e.g., carbon sequestration, water quality improvement, and regulation of air temperature and moisture), our study suggests that navigating the trade-off of water supply with these benefits in source watersheds is an important consideration in large water diversion projects." in line 457-460.

Given the potential feedbacks among vegetation, climate and human management, to avoid confusion and better reflect our finding, we changed our previous title "Vegetation Greening Significantly Reduced the Capacity of Water Supply to China's South-North Water Diversion Project" to be a moderate one: "Vegetation Greening Weakened the Capacity of Water Supply to China's South-North Water Diversion Project". Overall, we believe that the study now delivers a clear message about the hydrological effect of vegetation greening in the broader context of benefits that forests provide.

Technical comments:

- the green triangle for the Danjiangkou Reservoir in Figure 1 cannot be seen. In general: please avoid using red and green colours next to each other in figures as >10% of your readers may be red-green colour blind.

Response: Done as suggested. The triangle was changed as yellow.

[Figure]

- it will be very helpful for the reader to actually show the confidence intervals in the figures of the regression lines and/or trends, instead of only providing the numbers. For example in Figure 5, the reader can easily miss that the slope in the trend of the water yield is -2.9+/-10(!!!). Please also provide uncertainty intervals for the modeled water yield (Figure 4)

Response: The benefits of adding the confidence intervals on the regression lines come with detrimental visual effects. It will make some figures (4a, 5c) messy, and sometimes nearly impossible to follow. We added the confidence interval in Figure 3a, 5b, 6a, but did not in other figures.

**To Reviewer #2,**

With pleasure I read the revised manuscript by Zhang et al. on the effects of vegetation greening on water yield in the Upper Han River Basin. I was one of the reviewers of the first version of the manuscripts. The authors well answered to the questions and comments that were raised by the reviewers and incorporated the suggestions. The extra information in the methods section, and the short evaluation of the trend in the measured streamflow greatly increased the clarity of the study, and took away the questions and 'concerns' I had after reading the first version. Also the broader discussions of the effect of forest planting and feedbacks have added value to the manuscript.

Response: We appreciate your recognition of our revision. Your comments and suggestions helped a lot on improving the manuscript.

I have a few small suggestions that the authors could consider:

L203, "Here, the monthly drought index was calculated as the percentages of monthly WY to the mean WY of the same month during 2001-2018"? I assume this is calculated based on the modeled streamflow. Could you clarify this in the text?

Response: Clarified as "Here, the monthly drought index was calculated as the percentages of monthly WY to the mean WY of the same month during 2001-2018 based on simulated WY"

Figure 5: For clarity, I suggest to add to the caption of figure 5 that this figure contains the modeled results. For example: "Figure 5: modeled spatial and temporal variability of water yield in the Upper Han River Basin"

Response: Revised.

Figure 6: Since the change point detection was removed, you could remove the distinct orange coloring of WY/P ratio before 2003.

Response: Revised.